# Exogenous L-arginine increases intestinal stem cell function through CD90+ stromal cells producing mTORC1-induced Wnt2b

Qihang Hou[1], Yuanyang Dong[1], Jingxi Huang[1], Chaoyong Liao[1], Jiaqi Lei[1], Youli Wang[1], Yujiao Lai[1], Yifei Bian[1], Yang He[1], Jingjing Sun[1], Meng Sun[1], Qiuyu Jiang[1], Bo Wang[1], Zhengquan Yu [2], Yuming Guo[1] & Bingkun Zhang [1✉]

The renewal and repair of intestinal epithelium depend on the self-renewal of intestinal stem cells (ISCs) under physiological and pathological conditions. Although previous work has established that exogenous nutrients regulate adult stem cell activity, little is known about the regulatory effect of L-arginine on ISCs. In this study we utilize mice and small intestinal (SI) organoid models to clarify the role of L-arginine on epithelial differentiation of ISCs. We show that L-arginine increases expansion of ISCs in mice. Furthermore, CD90+ intestinal stromal cells augment stem-cell function in response to L-arginine in co-culture experiments. Mechanistically, we find that L-arginine stimulates Wnt2b secretion by CD90+ stromal cells through the mammalian target of rapamycin complex 1 (mTORC1) and that blocking Wnt2b production prevents L-arginine-induced ISC expansion. Finally, we show that L-arginine treatment protects the gut in response to injury. Our findings highlight an important role for CD90+ stromal cells in L-arginine-stimulated ISC expansion.

---

[1] State Key Laboratory of Animal Nutrition, Department of Animal Nutrition & Feed Science, College of Animal Science & Technology, China Agricultural University, Haidian District, Beijing 100193, China. [2] State Key Laboratories for Agrobiotechnology, College of Biological Sciences, China Agricultural University, Haidian District, Beijing 100193, China. ✉email: bingkunzhang@126.com

The mammalian intestinal tract is the primary organ for the digestion and absorption of nutrients in the body. Moreover, it is the first barrier of the body's defence system. The intestinal epithelium is the most active self-renewal tissue in mammals[1], being continuously renewed by self-renewal of intestinal stem cells (ISCs) harboured in the crypt bottom[2]. ISCs differentiate into progenitor cells, and these newly formed cells proliferate and differentiate along the crypt-villus axis of the intestine. Leucine-rich repeat containing G-protein coupled receptor 5 (Lgr5) has been identified as an important marker of active ISCs that can generate differentiated epithelial cell types over long periods of time[3]. A second population of quiescent "reserve" ISCs are located at the so-called '+4' position[4]. ISCs can differentiate into multiple cell types. Although the major intestinal epithelial cells are absorptive enterocytes, the intestinal epithelium also contains secretory cell lineages, including Paneth cells, which support the ISC niche and secrete antimicrobial peptides; mucus-producing goblet cells; various hormone-secreting enteroendocrine cells; M cells and tuft cells[2]. The function of ISCs is tightly regulated by a variety of signalling pathways to maintain gut homeostasis. Among the modulators of ISCs in crypt niches, Wnt/β-catenin signalling is indispensable for ISC proliferation and differentiation[1]. Recent studies confirm that one of the components comprising the ISC niches are secretory epithelial cells adjacent to the ISCs—Paneth cells in the small intestine[5] and cKit+[6] or Reg4+[7] cells in the colon—that offer essential factors for ISC maintenance, such as epidermal growth factor, Wnt3, and DLL1/4[8–10]. Moreover, many factors produced by stromal cells are indispensable for the maintenance of ISCs, such as Wnt2b[11], the Lgr4/5 ligand R-spondin1[12–14], and Gremlin1[15,16].

Many factors, such as pathogen infection, malnutrition, and even chemotherapy, cause damage to the intestinal barrier and induce intestinal inflammation, including inflammatory bowel disease (IBD) and intestinal mucositis[17–19]. Nutritional interventions may affect gastro-intestinal mucositis, and amino acids, such as L-arginine, may prevent mucositis-associated intestinal alterations[20]. 5-FU is widely used in various cancers treatment, including cancers of the aerodigestive tract[21]. However, 5-FU treatment causes gastrointestinal mucositis in as many as 80% of patients[22]. Previous studies indicated that dietary L-arginine supplementation could reduce chemotherapy-induced intestinal mucosal injury in mice[23–25]. L–arginine supplementation attenuates the degree of tissue damage in intestinal ischaemia and promotes healing of the intestinal mucosa[26]. L-arginine also participates in wound healing and intestinal epithelial cell proliferation under intestinal physiology[27,28].

Recent studies show that dietary nutrients play an important role in maintaining tissues and adult stem cells in various tissues[29]. For example, high-fat diets enhance ISC function through activation of a peroxisome proliferator-activated receptor-delta programme[30]. The mechanistic target of mTORC1 is a key mediator of eukaryotic growth that coordinates anabolic and catabolic cellular processes with inputs such as growth factors and nutrients, including amino acids[31–33]. In mammals, L-arginine is an extremely important amino acid that promotes diverse biological processes, such as protein synthesis, muscle growth, and immune cell activation, largely mediated through activation of mTORC1[34–36].

The majority of published papers concerning the effects of L-arginine on the intestinal barrier have focused on the regulation of tight junctions and maintenance of intestinal morphology through mTORC1 signalling pathway[28,37,38]. However, whether L-arginine plays a role in the proliferation of ISCs is still unclear. To elucidate the role of L-arginine in intestinal proliferation and maintenance of the intestinal mucosal barrier, we utilized a mouse model and intestinal organoid models to clarify the role of L-arginine on ISC-mediated intestinal epithelial renewal. Our findings demonstrated that regulation of the mTORC1 pathway with respect to CD90+ stromal cells by exogenous L-arginine mediates ISC function in a Wnt2b/β-catenin pathway-dependent manner. We also establish a model in which the mammalian ISC niche couples organismal nutrient levels to stem cell function.

## Results

**L-arginine supplementation promotes ISC-mediated intestinal regeneration**. To assess the efficacy of L-arginine on intestinal homeostasis, 8-week-old male Lgr5-eGFP-IRES-creERT2 reporter mice, hereinafter referred to as Lgr5-GFP[3], were treated with normal water or 7 mg/ml L-arginine in their drinking water. Consistent with previous publications[39,40], mice were orally administrated L-arginine supplementation (1.5 g/kg body weight/day) for 2 weeks. According to the daily water consumption by mice, we calculated that the concentration of L-arginine added to the drinking water was about 7 mg/ml. L-arginine concentration was higher in SI content, SI tissue, and serum after L-arginine treatment (Fig. 1b). No changes in daily weight gain were observed (Supplementary Fig. 1a). In mice fed L-arginine, the small intestine was morphologically normal (Fig. 1c), with increases in SI length and mass (Supplementary Fig. 1b). Moreover, mice fed L-arginine had increased SI crypt height and villus height compared to the control group (Fig. 1c). Consistently, we observed a higher percentage of 5-ethynyl-2′-deoxyuridine (EdU)-positive cells in SI crypts from mice fed L-arginine (Fig. 1d), indicating that L-arginine promoted intestinal epithelial cell proliferation in crypt areas. To address how L-arginine supplementation influenced the frequency of ISCs, we performed immunofluorescence staining for olfactomedin 4 (Olfm4), a marker that is co-expressed by Lgr5+ ISCs[41]. L-Arginine supplementation led to increased Olfm4+ ISCs compared to control mice (Fig. 1f). We also performed flow cytometry for live Lgr5hi cells (known as Lgr5+ ISCs)[42] in SI crypts. L-arginine supplementation increased numbers of Lgr5+ in SI crypts compared to the control group (Fig. 1e). Notably, L-arginine supplementation also induced increased lysozyme+ Paneth cells (Supplementary Fig. 1c) but mildly reduced Muc2+ goblet cells in SI crypts (Supplementary Fig. 1d). Because L-arginine supplementation increased the frequency of ISCs, we examined whether it also promoted ISC-mediated intestinal epithelium regeneration. We isolated crypts from the small intestine and examined their potential to form organoid bodies in vitro. Crypts from mice fed L-arginine were more likely to form organoid bodies than those from normal fed controls (Fig. 1g). Moreover, we tested the potential of sorted Lgr5+ ISCs from Lgr5-GFP mice to form clonal organoid bodies. ISCs from Lgr5-GFP mice treated with L-arginine were more likely to form organoid bodies than those from the control group (Fig. 1h). Collectively, these findings demonstrated that L-arginine supplementation promotes ISC-mediated intestinal regeneration.

**L-arginine supplementation has no direct effect on Lgr5+ ISC function**. The enhanced regenerative activity of ISCs in mice fed L-arginine next prompted us to ask how exogenous L-arginine affects the frequency and function of ISCs. To answer this, we isolated live Lgr5+ ISCs from Lgr5-GFP mice by fluorescence-activated cell sorting (FACS)[42]. We investigated the organoid-forming capacity of Lgr5+ ISCs cultured in ENR-media (epidermal growth factor, noggin, and R-spondin 1) supplemented with L-arginine (1 mM) or not. Interestingly, treatment with L-arginine increased neither numbers of Lgr5+ ISCs nor organoid-forming capacity of Lgr5+ ISCs for 3 and 9 days

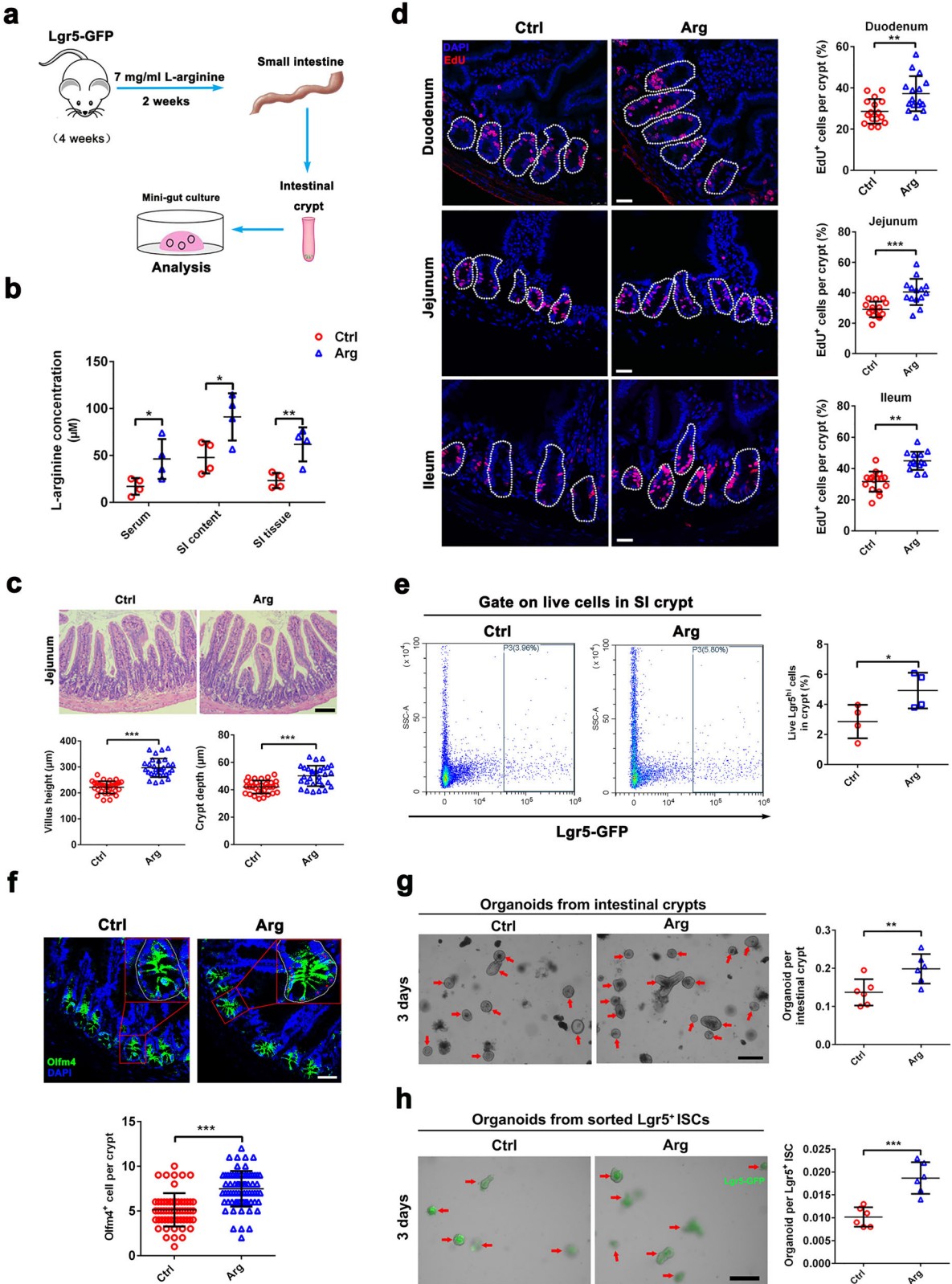

of cultivation compared to groups treated with ENR alone (Fig. 2b–e). Consistently, the number of EdU⁺Lgr5⁺ cells of SI organoids derived from ISCs exhibited no change between the two groups (Fig. 2f, g). Interestingly, we observed that high doses of L-arginine supplementation (2 mM, 3 mM) could even destroy intestinal organoids (Supplementary Fig. 2a). We also assessed

the mRNA expression of markers for both Lgr5⁺ ISCs (*Lgr5*, *Olfm4*) and +4 quiescent ISCs (*Bmi1*, *mTert*) in organoids treated with L-arginine (1 mM) or not by RT-PCR (Supplementary Fig. 2b). There were no differences between the two groups. These results indicated that L-arginine supplementation has no direct effect on Lgr5⁺ ISC function.

**Fig. 1 L-arginine supplementation promotes ISC-mediated intestinal regeneration. a–h** Mice were treated with normal water or 7 mg/ml L-arginine in their drinking water for 14 days. **a** Experimental schedule for analysis of crypt and ISC function in mice treated with/without L-arginine. **b** Measurement of L-arginine concentration in SI content, SI tissues, and serum of mice treated with L-arginine or not, $n = 4$ mice. **c** The SI villus height and crypt depth by H&E staining, $n = 6$ mice. Scale bar, 50 μm. **d** Immunostaining of EdU (red) and DAPI (blue) in small intestine (white dotted line marks crypt area). The number of EdU+ cell per crypt was counted, $n = 3$ mice per group. Scale bar, 25 μm. **e** Flow Cytometry analysis of live Lgr5hi ISC (Lgr5-GFP) frequency in SI crypt, $n = 3$ mice. **f** Olfm4 (green) and DAPI (blue) in SI crypts (red asterisk marks Olfm4+ ISCs). The number of Olfm4+ cell per crypt was counted, $n = 3$ mice. Scale bar, 50 μm. **g** SI organoid frequency of crypts from mice treated with/without L-arginine. Representative images of crypt culture from each group are shown (red arrow marks organoids), $n = 6$ mice. **h** Organoid formation per Lgr5+ ISCs sorted from mice treated with/without L-arginine, $n = 6$ mice. A representative image of organoids at day 3 is shown (red arrow marks organoids). Scale bar, 200 μm. Data are the mean ± SD; comparisons performed with t-tests (two groups) or analysis of variance (ANOVA) (multiple groups). *$P < 0.05$, **$P < 0.01$, ***$P < 0.001$. Results are representative of two or three independent experiments.

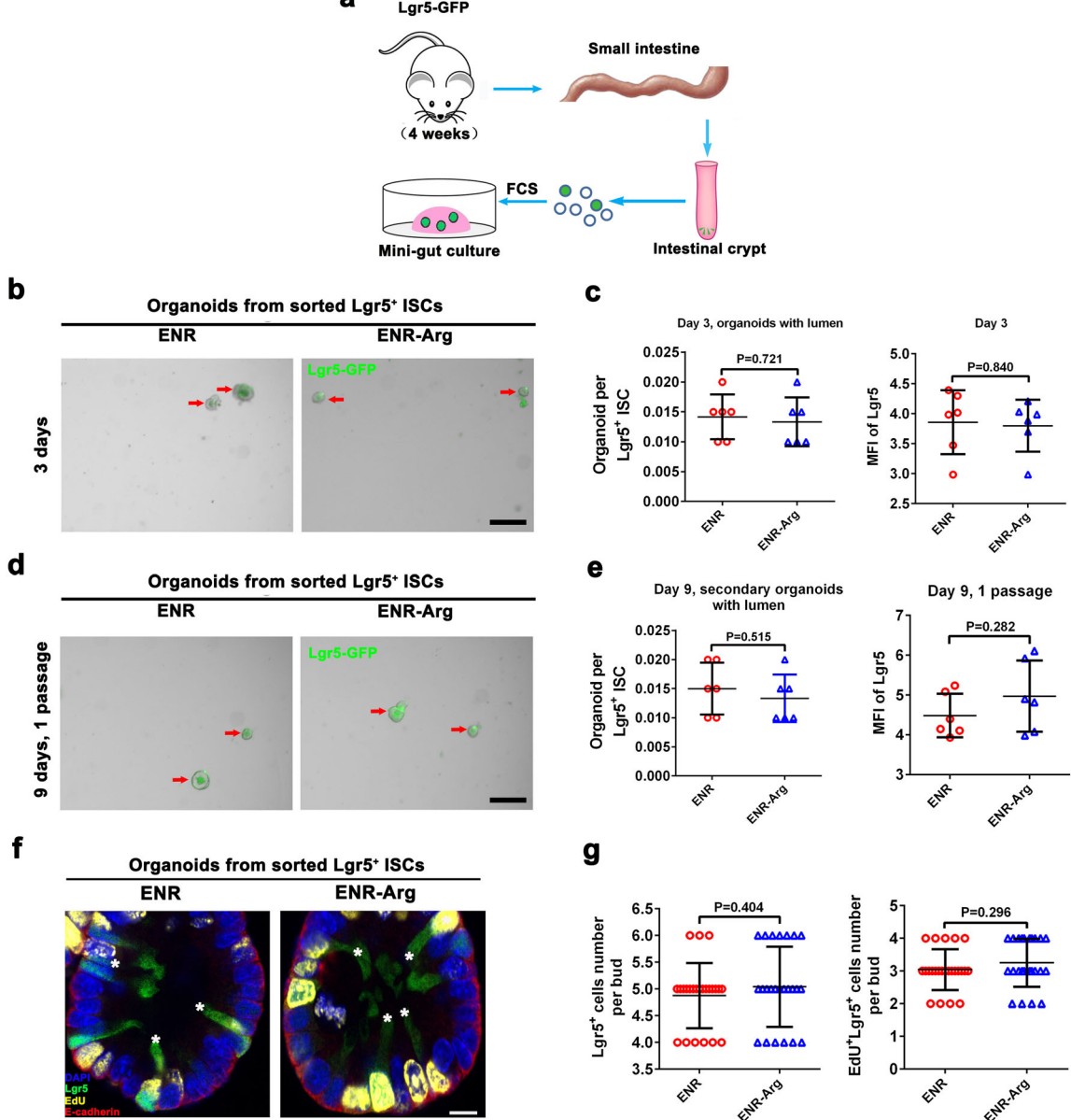

**Fig. 2 L-arginine supplementation has no direct effect on Lgr5+ ISC function.** Lgr5+ ISCs were cultured in ENR-medium or ENR-medium supplemented with 1 mM L-arginine. **a** Experimental schedule for mini-gut culture of Lgr5+ ISCs in vitro. **b** Organoid formation per Lgr5+ ISCs treated with/without L-arginine, $n = 6$. A representative image of organoids at day 3 is shown (red arrow marks organoids). Scale bar, 200 μm. **c** Quantitation for Lgr5 was measured by MFI, $n = 6$. **d** Organoid formation per Lgr5+ ISCs derived from dissociated organoids in Fig. 2b, $n = 6$. A representative image of organoids at day 9 is shown (red arrow marks organoids). Scale bar, 200 μm. **e** Quantitation for Lgr5 was measured by MFI, $n = 6$. **f** Immunostaining of Lgr5-GFP (green), EdU (yellow), E-cadherin (red) and DAPI (blue) in the SI organoids (white asterisk marks EdU+Lgr5+ ISCs). Scale bar, 10 μm. **g** The number of Lgr5+ cell and Lgr5+EdU+ cells per bud was counted, $n = 24$ buds per group. Data are the mean ± SD; comparisons performed with t-tests (two groups) or analysis of variance (ANOVA) (multiple groups). *$P < 0.05$, **$P < 0.01$, ***$P < 0.001$. Results are representative of two or three independent experiments.

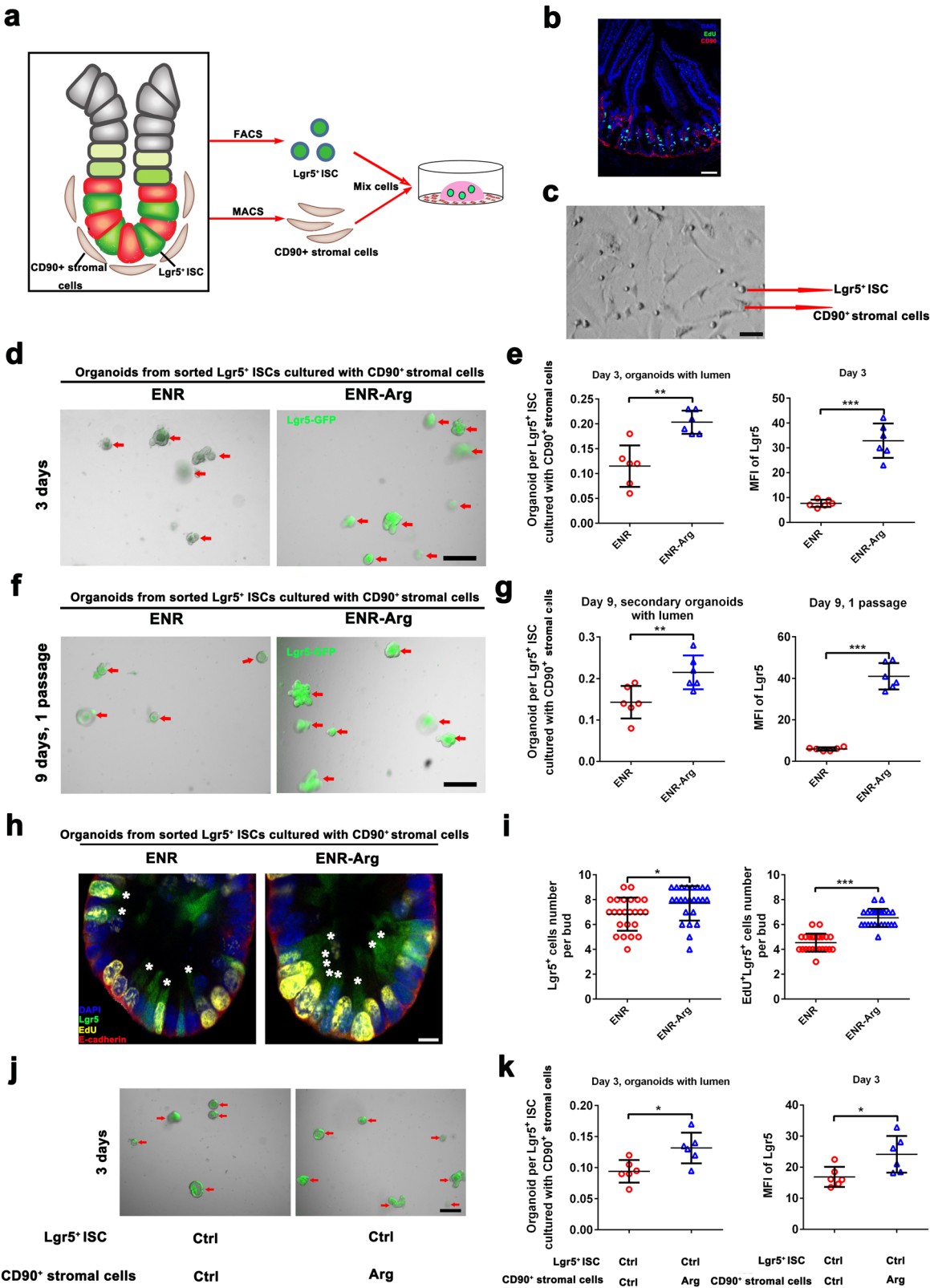

**CD90[+] stromal cells support L-arginine-mediated Lgr5[+] ISC regeneration.** Previous studies demonstrated that Paneth cells provide important factors for ISC maintenance[5], such as DLL1/4, TGF-α, and Wnt3[8–10]. The enhanced regenerative activity of ISCs in mice fed L-arginine led us to ask whether ISCs responded to L-arginine through Paneth cell niches. However, exogenous L-arginine treatment had no effect on numbers of lysozyme[+]

Paneth cell in organoids (Supplementary Fig. 2c). Similar results were further verified by the mRNA expression of Paneth cell marker genes (*Lyz1*, *Defa6*) (Supplementary Fig. 2d). Moreover, we found that L-arginine supplementation had no impact on mRNA expression of *Wnt2b*, *Wnt3*, *Axin2*, or *Ctnnb1* (Supplementary Fig. 2e). Collectively, these results indicated that the effects of exogenous L-arginine treatment on ISC function might

**Fig. 3 CD90[+] stromal cells support L-arginine-mediated Lgr5[+] ISC regeneration.** Lgr5[+] ISC-CD90[+] stromal cells co-culture model was cultured in ENR-medium or ENR-medium supplemented with 1 mM L-arginine. **a** Experimental schedule for the co-culture model of Lgr5[+] ISCs and CD90[+] stromal cells. **b** Immunostaining of EdU (green), CD90[+] (red), and DAPI (blue) in the jejunum. Scale bar, 50 μm. **c** Lgr5[+] ISCs cultured with CD90[+] stromal cells were observed with a light microscope. Scale bar, 10 μm. **d** Organoid formation per Lgr5[+] ISCs treated with/without L-arginine in co-culture model, n = 6. A representative image of organoids at day 3 is shown (red arrow marks organoids). Scale bar, 200 μm. **e** Quantitation for Lgr5 was measured by MFI, n = 6. **f** Organoid formation per Lgr5[+] ISCs derived from dissociated organoids in Fig. 3d, n = 6. A representative image of organoids at day 9 is shown (red arrow marks organoids). Scale bar, 200 μm. **g** Quantitation for Lgr5 was measured by MFI, n = 6. (**h**) Immunostaining of Lgr5-GFP (green), EdU (yellow), E-cadherin (red) and DAPI (blue) in the SI organoids (white asterisk marks EdU[+]Lgr5[+] ISCs). Scale bar, 5 μm. **i** The number of Lgr5[+] cell and Lgr5[+]EdU[+] cells per bud was counted, n = 24 buds. **j** Organoid formation per Lgr5[+] ISCs cultured with CD90[+] stromal cells derived from mice treated with/without L-arginine, n = 6. A representative image of organoids at day 3 is shown (red arrow marks organoids). Scale bar, 200 μm. **k** Quantitation for Lgr5 was measured by MFI, n = 6. Data are the mean ± SD; comparisons performed with t-tests (two groups) or analysis of variance (ANOVA) (multiple groups). *P < 0.05, **P < 0.01, ***P < 0.001. Results are representative of two or three independent experiments.

not be mediated through the Paneth cells niche. Recent studies demonstrated that a number of factors produced by intestinal stromal cells have an essential role in the maintenance of ISCs[11,43]. A recent study reported that CD90[+] stromal cells are located at the base of crypts and support intestinal epithelial growth[44]. In our study, we found that CD90 was broadly expressed in stromal cells adjacent to ISCs (Fig. 3b). The enhanced regenerative activity of ISCs in mice fed L-arginine led us to examine whether ISCs responded to L-arginine through the stromal cell niches. To test this, we sorted Lgr5[+] ISCs and CD90[+] stromal cells from Lgr5-GFP mice and built an ISC-stromal cell co-culture model and assayed their ability to form organoid bodies in culture (Fig. 3c). As shown in Fig. 2b–e, very few Lgr5[+] ISCs established organoid bodies on their own, but, when co-cultured with CD90[+] stromal cells, more than 10% of ISCs generated organoid bodies (Fig. 3d–g), indicating that CD90[+] stromal cells have an essential role in the maintenance of ISCs. Notably, Lgr5[+] ISCs cultured in ENR-L-Arg medium were more likely than those cultured in ENR medium to promote organoid body formation when co-cultured with CD90[+] stromal cells (Fig. 3d, e). The effects of L-arginine on SI organoids were also consistent with the proliferation status of ISCs. Not only did L-arginine supplementation promote primary organoid body formation, but these organoids also gave rise to more and larger secondary organoid bodies, even when individually subcloned (Fig. 3f, g). Notably, we observed higher quantities of Lgr5[+] and EdU[+]Lgr5[+] cells in SI organoids treated with L-arginine in the co-culture model (Fig. 3h, i). To further solidify the conclusion that the effects of exogenous L-arginine on ISCs primarily originate from the stromal population, we sorted and mixed CD90[+] stromal cells from L-arginine treated mice with non-treated Lgr5[+] ISCs and assayed their ability to form organoid bodies in vitro (Fig. 3j). ISCs co-cultured with stromal cells from L-arginine treated mice generated more organoids (Fig. 3k). Overall, utilizing the ISC-stromal cell co-culture model, we observed that CD90[+] stromal cells support L-arginine-mediated Lgr5[+] ISC regeneration.

**L-arginine supplementation stimulates intestinal epithelial regeneration through the Wnt/β-catenin pathway.** Among the modulators of ISCs in crypt niches, Wnt/β-catenin signal is indispensable for ISC proliferation and differentiation[1]. We found that L-arginine supplementation increased expression of active β-catenin in SI organoids cultured with CD90[+] stromal cells, which was confirmed by immunofluorescence staining (Fig. 4a) and western blot (Fig. 4b). Meanwhile, L-arginine supplementation enhanced expression of Wnt/b-catenin pathway-related genes (Ctnnb1, Axin2, and GSK3b) in SI organoids derived from ISCs cultured with CD90[+] stromal cells (Fig. 4c). We further confirmed in vivo that the increased expression of active β-catenin occurred in the SI crypts of mice fed L-arginine, unlike

control mice (Fig. 4d). Meanwhile, L-arginine enhanced the gene expression of Ctnnb1 and Axin2 in the SI crypts (Fig. 4e). However, when CD90[+] stromal cells were absent, L-arginine had no effect on β-Catenin expression in SI organoids (Supplementary Fig. 2a). Similar results were verified by mRNA expression of GSK3β, Axin2, and Ctnnb1 (Supplementary Fig. 2b). These results indicated that L-arginine supplementation stimulates intestinal epithelial regeneration through the Wnt/β-catenin pathway.

**L-arginine-induced Wnt2b secretion from CD90[+] stromal cells supports ISC-mediated intestinal epithelial renewal.** Recent studies demonstrated that many factors produced by stromal cells are indispensable for the maintenance of ISCs, such as Wnt2b[11], the Lgr4/5 ligand R-spondin1[12–14], and Gremlin1[15,16]. In our study, we found that L-arginine supplementation increased expression of Wnt2b in CD90[+] stromal cells, whereas levels of other Wnts (Wnt1, Wnt3, Wnt3a, Wnt5a, and Wnt6), R-spondin1, and Gremlin1) did not change (Fig. 5a). Notably, expression levels of Wnt2b in CD90[+] stromal cells were highly activated after addition of L-arginine in vitro (Fig. 5b, c). We further confirmed in vivo that L-arginine increased expression of Wnt2b in SI crypts (Fig. 5d, e). We next examined whether Wnt2b affected the function of ISCs by sorting Lgr5[+] ISCs and assaying their ability to form organoid bodies in culture. Notably, Lgr5[+] ISCs cultured in ENR-Wnt2b medium were more likely than those cultured in ENR medium to promote organoid body formation (Fig. 5f). Furthermore, treatment with Wnt2b antibody (Wnt2b-Ab) inhibited organoid body formation of Lgr5[+] ISCs promoted by L-arginine in the co-culture model (Fig. 5g). To test the specificity of L-arginine's effects on Wnt2b-mediated secretion of CD90 stromal cells, the medium of CD90[+] stromal cells was supplemented with 1 mM different amino acids (L-arginine, L-leucine, L-lysine, or L-glutamine), respectively (Supplementary Fig. 4a). We detected the concentration of WNT2B protein in the culture supernatant at day 3 by ELISA (Supplementary Fig. 4b). Interestingly, both L-arginine and L-leucine up-regulated expression of WNT2B protein. However, L-lysine and L-glutamine had no effects. These results indicated that increased expression of Wnt2b in CD90[+] stromal cells mediates the effects of L-arginine on ISC function.

**L-arginine supplementation stimulates Wnt2b synthesis in CD90[+] stromal cells through the mTORC1 signalling pathway.** Cationic amino acid transporter (CAT, Slc7a), known as system y[+], transports cationic amino acids, such as L-arginine, L-lysine, L-histidine, and L-ornithine[45]. In our study, we found that L-arginine increased expression of Slc7a1 and Slc7a4 in CD90[+] stromal cells (Fig. 6a, b), whereas levels of other Slc7as (Slc7a2, Slc7a3, Slc7a6, Slc7a7 and Slc7a9) were unchanged (Supplementary Fig. 5). L-arginine is extremely important in diverse

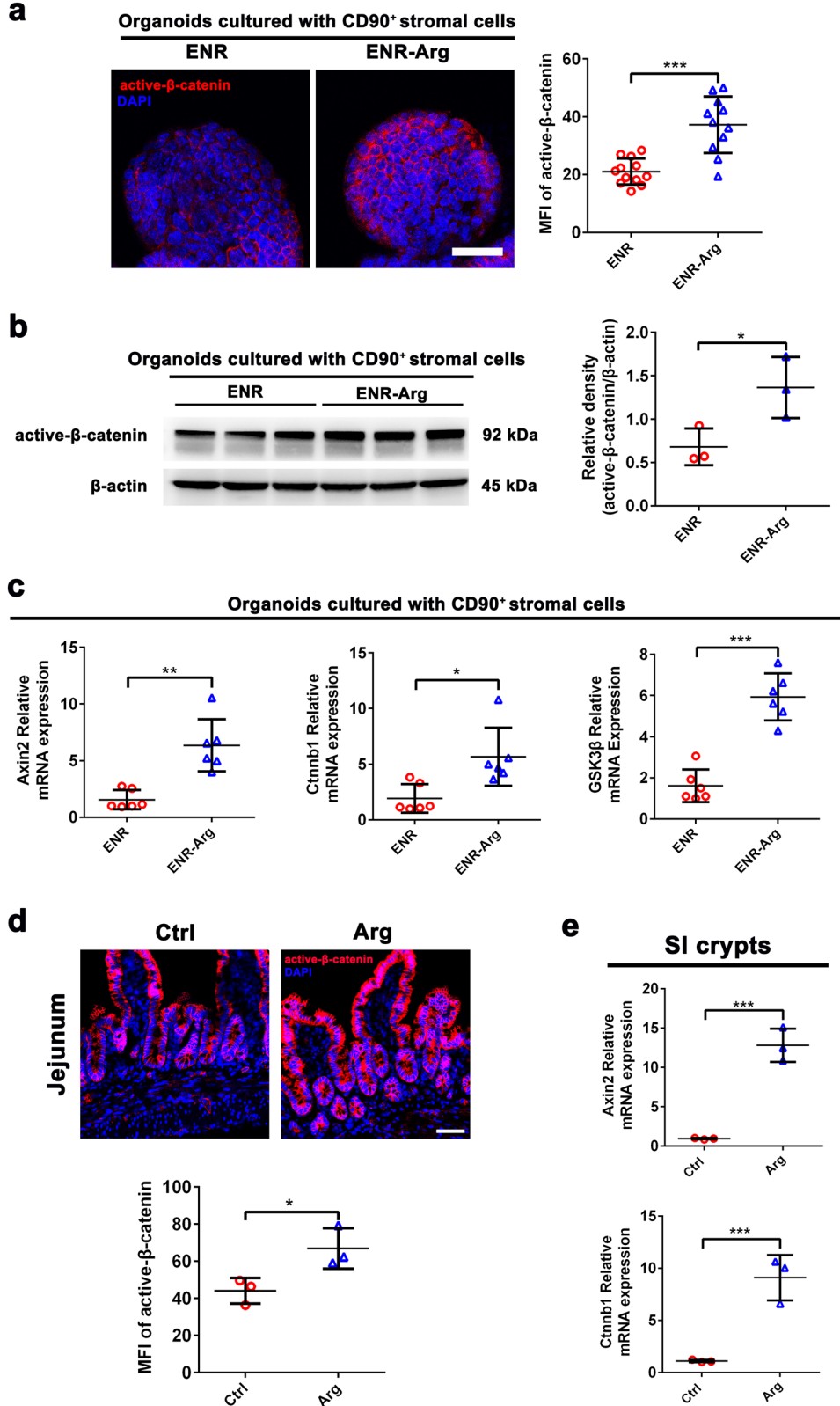

functions, such as protein synthesis, which are largely mediated through activation of mTORC1[36]. Expression of total and phosphorylated mTOR, p70S6K, and 4EBP1 were readily detected in CD90+ stromal cells treated with L-arginine or not. Notably, L-arginine supplementation increased the abundance of phosphorylated forms of mTOR, p70S6K and 4EBP1 (Fig. 6c, d). In addition, expression levels of Wnt2b in CD90+ stromal cells were highly activated in response to the addition of L-arginine and the mTOR activator MHY1485 in vitro (Fig. 6e, f). Furthermore, treatment with the mTOR inhibitor Rapamycin inhibited Wnt2b expression promoted by L-arginine and MHY1485 in CD90+ stromal cells (Fig. 6e, f). These results indicated that L-arginine supplementation stimulates Wnt2b synthesis in CD90+ stromal cells through the mTORC1 signalling pathway.

**Fig. 4 L-arginine supplementation stimulates intestinal epithelial regeneration through the Wnt/β-catenin pathway.** Lgr5[+] ISC-CD90[+] stromal cells co-culture model was cultured in ENR-medium or ENR-medium supplemented with 1 mM L-arginine for 72 h, respectively. **a** Immunostaining of active β-catenin (red) and DAPI (blue) in the SI organoids cultured with CD90[+] stromal cells. Scale bar, 10 μm. Quantitation for active β-catenin was measured by MFI, n = 12 organoids. **b** Nuclear protein levels of β-actin and β-catenin were measured by western blotting assay in the SI organoids cultured with CD90[+] stromal cells, n = 3. **c** qPCR of relative mRNA expression of *Ctnnb1*, *Axin2*, and *GSK3β* genes of the Wnt/β-catenin axis in SI organoids cultured with CD90[+] stromal cells. Expression show is relative to *GAPDH* gene, n = 6. **d** Immunostaining of active β-catenin (red) and DAPI (blue) in jejunal crypts. Quantitation for active β-catenin was measured by MFI, n = 3 mice. **e** qPCR of relative mRNA expression of *Ctnnb1* and *Axin2* in SI crypts, n = 3. Data are the mean ± SD; comparisons performed with *t*-tests (two groups) or analysis of variance (ANOVA) (multiple groups). *P < 0.05, **P < 0.01, ***P < 0.001. Results are representative of two or three independent experiments.

**L-arginine supplementation protects the gut from 5-FU and TNF-α induced intestinal epithelial damage in a Wnt2b-dependent manner.** To further clarify the protective role of L-arginine and address clinical indices, we treated mice with 5-FU to destroy the SI epithelium as previously described[21]. Dietary L-arginine supplementation before 5-FU treatment protected mice from the destruction of the intestinal mucosal (Fig. 7a). Pre-feeding of L-arginine protected mice from loss of Lgr5[+] ISCs due to treatment with 5-FU as well (Fig. 7b). We next obtained SI crypts from each group of mice. Then, we manipulated organoids (400 crypts/well) and counted the number of budding-formed organoids. Most importantly, feeding L-arginine before treatment with 5-FU resulted in more budding-formed organoids than were in the control group (Fig. 7c). To test whether the protective effect of L-arginine is mediated through a Wnt2b-dependent manner, we neutralized WNT2B protein in vivo. Treatment of Wnt2b neutralizing antibody reversed the protective effect of L-arginine on intestinal mucosal damage and loss of Lgr5[+] ISCs (Fig. 7a, b). Meanwhile, Wnt2b neutralizing antibody inhibited the stimulatory effect of L-arginine treatment on crypt-formed organoids in mice treated with 5-FU (Fig. 7c). Moreover, we treated SI organoids with murine TNF-α to destroy the SI epithelium as previously described[46]. TNF-α successfully induced SI organoid disruption in a co-culture model. However, L-arginine treatment alleviated and protected cells from damage induced by TNF-α on SI organoids, allowing cells to maintain normal morphology (Fig. 7d). Notably, WNT2B protein treatment protected cells from damage induced by TNF-α on SI organoids in a co-culture model (Fig. 7e). Moreover, the protective effect of L-arginine and WNT2B protein on the morphology of SI organoids damaged by TNF-α was inhibited by the addition of a Wnt2b neutralizing antibody (Fig. 7e). Overall, it is evident that L-arginine supplementation protects the gut from 5-FU and TNF-α induced intestinal epithelial damage in a Wnt2b-dependent manner.

## Discussion

Recent studies indicate that dietary nutrients play an important role in the maintenance of intestinal tissues and stem cells[29,47–50]. In this study, we utilized mice and SI organoid models to clarify the role of L-arginine on epithelial differentiation of ISCs. Here, we demonstrated that the L-arginine supplementation increased expansion of ISCs in mice. By constructing a co-culture model, we found that CD90[+] stromal cells, a key constituent of mammalian ISC niches, augmented stem cell function in response to L-arginine. Moreover, increased expression of Wnt2b in CD90[+] stromal cells—a ligand of Wnt/β-catenin signalling pathway—mediated the effects of L-arginine on ISC function. We found that L-arginine stimulates Wnt2b secretion of CD90[+] stromal cells through mTORC1 signals. Pre-treatment with L-arginine accelerated ISC-mediated intestinal epithelial regeneration and protected the gut in response to injury provoked by murine TNF-α and the chemotherapeutic 5-FU. Our findings establish that regulation of mTORC1 pathway in CD90[+] stromal cells by exogenous L-arginine mediates ISC function in a

Wnt2b/β-catenin pathway-dependent manner. We also established a model in which the mammalian ISC niches couples organismal nutrient levels to stem cell function (Fig. 8).

A previous study demonstrated that pre-treatment with L-arginine after mucosal injury resulted in a higher survival rate and decreased mucosal barrier permeability in rats[51]. Meanwhile, L-arginine supplementation improved the clinical parameters in colitis induced by DSS[28,52]. These investigations indicated that L-arginine might represent a potential therapy for intestinal mucosal injury. Mammalian ISCs maintain intestinal homeostasis by undergoing either self-renewal or differentiation divisions that generate more stem cells or restricted progenitors, respectively[53]. Recent studies indicate that dietary nutrients play an important instructive role in the maintenance of tissues and adult stem cells in diverse tissues[29]. These findings raise the question of whether L-arginine mediates these effects on ISCs, and whether the mammalian stem cell niche is involved. To address how exogenous L-arginine influences the frequency of ISCs, we used Lgr5-GFP mice, which enable identification and isolation of Lgr5[+] ISCs[3]. Interestingly, pre-treatment with L-arginine increased the number of Lgr5[+] ISCs, Paneth cells, and EdU[+] cells in SI crypts, indicating that output and migration into the villi from this compartment may be increased. We next investigated whether L-arginine also promotes regeneration of the SI epithelium. We tested the potential of isolated SI crypts to form clonal, multipotent organoid bodies in vitro[54]. Crypts from mice fed L-arginine were more likely to form organoid bodies than those from controls.

In light of these results, we next examined how exogenous L-arginine affects the frequency and function of ISCs. To test this, we isolated Lgr5[+] ISCs from Lgr5-GFP mice by FACS[42]. We investigated the organoid-forming capacity of Lgr5[+] ISCs cultured in ENR-media supplemented with L-arginine or not. Interestingly, treatment with L-arginine neither increased numbers of Lgr5[+] ISCs nor the organoid-forming capacity of Lgr5[+] ISCs organoid when compared to groups treated with ENR alone. These results indicated that L-arginine supplementation does not directly affect Lgr5[+] ISC function. Both intestinal crypts and ISCs isolated from L-arginine-treated mice were more likely to form organoid bodies than those from the control group. L-arginine supplementation promoted not only crypts but also ISC function. However, L-arginine supplementation did not affect the function of ISCs sorted from untreated mice in vitro. Metabolism in the stem cell niche is quite complex. Perhaps there is some metabolic memory in ISCs sorted from L-arginine-treated mice. Future studies are needed to further explore the complexity in understanding how metabolism regulates stem cell function and regeneration through ISC niche factors.

Recent studies demonstrated that intestinal stromal cells[55], Paneth cells[5], and intestinal immune cells[56,57] have essential roles in the maintenance of ISCs. A recent study reported that CD90[+] stromal cells are located at the base of crypts and support intestinal epithelial growth[44]. We next investigated whether ISCs responded to L-arginine through the Paneth cell niches. However,

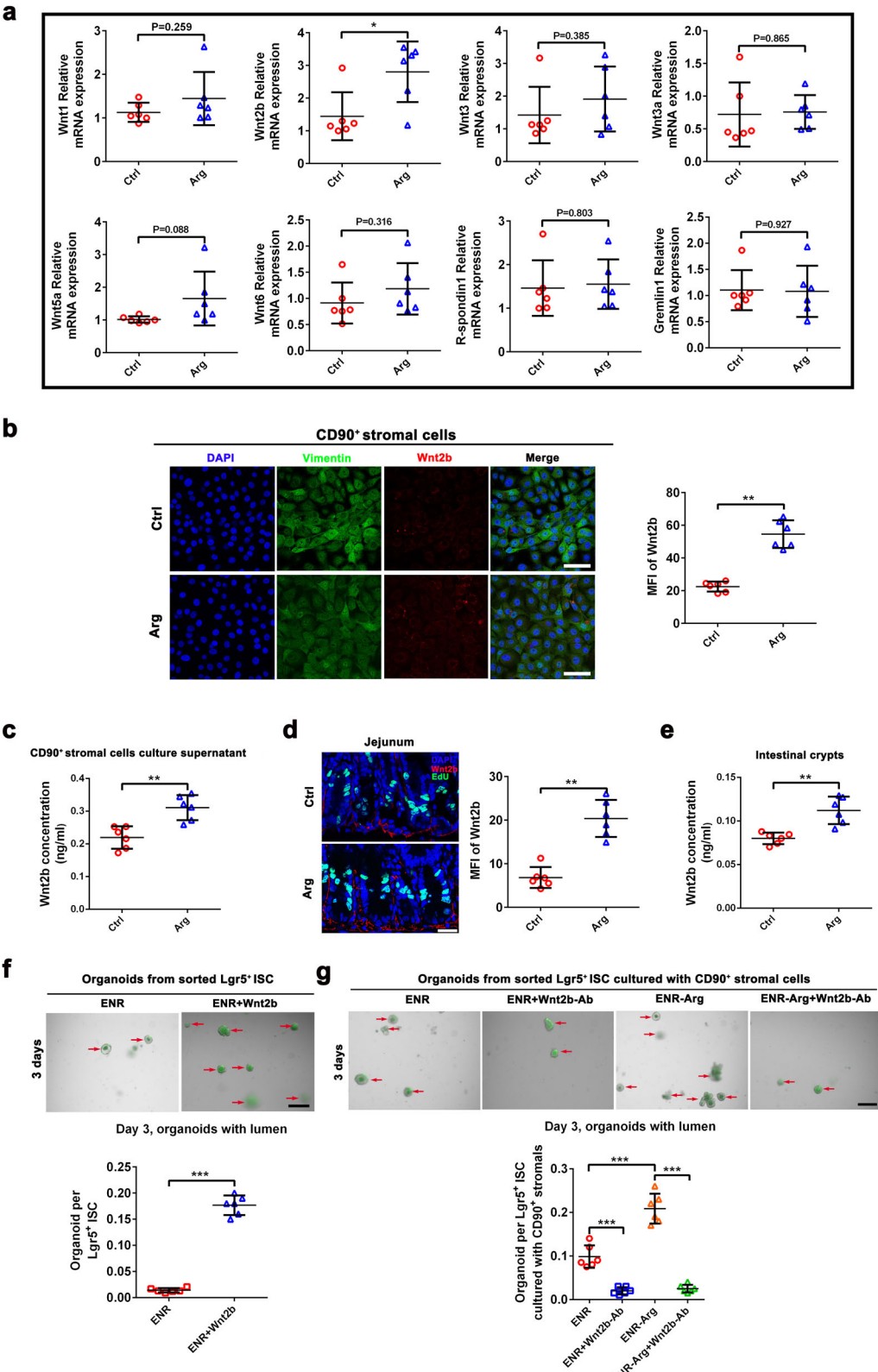

exogenous L-arginine treatment had no effect on numbers of Paneth cells in the organoids. Moreover, we found that L-arginine supplementation did not affect mRNA expression of *Wnt2b*, *Wnt3*, *Axin2*, or *Ctnnb1*. Collectively, these results indicated that the effects of exogenous L-arginine treatment on ISC function might not be mediated through the Paneth cells niche—or at least not primarily. Interestingly, we found that CD90 is broadly

expressed in the stromal cells adjacent to ISCs. To test whether ISCs respond to L-arginine through the stromal cells niche, we built an Lgr5[+] ISC-CD90[+] stromal cell co-culture model and assayed their ability to form organoid bodies. Very few Lgr5[+] ISCs established organoid bodies on their own; however, when co-cultured with CD90[+] stromal cells, large, increased numbers of ISCs were generated, indicating that CD90[+] stromal cells play

**Fig. 5 L-arginine-induced Wnt2b secretion from CD90+ stromal cells supports ISC-mediated intestinal epithelial renewal.** CD90+ stromal cells was cultured in medium supplemented with L-arginine (1 mM) or not for 72 h respectively. **a** qPCR of relative mRNA expression of *Wnt2b, Wnt1, Wnt3, Wnt3a, Wnt5a, Wnt6, R-spondin 1,* and *Gremlin1* genes in CD90+ stromal cells. Expression show is relative to *GAPDH* gene, n = 6. **b** Immunostaining of Wnt2b (red), Vimentin (green), and DAPI (blue) in CD90+ stromal cells. Scale bar, 10 μm. Quantitation for Wnt2b was measured by MFI, n = 6. **c** The concentration of Wnt2b in CD90+ stromal cells culture medium was detected by ELISA, n = 6. **d** Immunostaining of Wnt2b (red), EdU (green) and DAPI (blue) in jejunum. Scale bar, 50 μm. Quantitation for Wnt2b was measured by MFI, n = 6. (**e**) The concentration of Wnt2b in the SI crypts was detected by ELISA, n = 6. **f** Organoid formation per Lgr5+ ISCs treated with/without mouse recombinant WNT2B protein (300 ng/ml), n = 6. A representative image of organoids at day 3 is shown (red arrow marks organoids). Scale bar, 200 μm. **g** Organoid formation per Lgr5+ ISCs treated with mouse recombinant WNT2B protein (100 ng/ml), L-arginine (1 mM), Wnt2b antibody (1 μg/ml), and their combinations in co-culture model, n = 6. A representative image of organoids at day 3 is shown (red arrow marks organoids). Scale bar, 200 μm. Data are the mean ± SD; comparisons performed with t-tests (two groups) or analysis of variance (ANOVA) (multiple groups). *P < 0.05, **P < 0.01, ***P < 0.001. Results are representative of two or three independent experiments.

an essential role in the maintenance of ISCs. Notably, Lgr5+ ISCs were more likely to promote organoid body formation when treated with L-arginine in the co-culture model. Overall, utilizing the ISC-stromal cell co-culture model, we found that CD90+ stromal cells supported L-arginine-mediated Lgr5+ ISC regeneration.

Recent studies demonstrated that a number of factors produced by intestinal stromal cells have an essential role in the maintenance of ISCs. In our study, we found that L-arginine increased expression of Wnt2b in CD90+ stromal cells. We next examined whether Wnt2b affects the function of ISCs. To test this, we sorted Lgr5+ ISCs and assayed their ability to form organoid bodies in culture. Notably, Lgr5+ ISCs cultured in ENR-Wnt2b medium were more likely than those cultured in ENR medium to promote organoid body formation. These results indicated that increased expression of Wnt2b in CD90+ stromal cells—a ligand of Wnt/β-catenin signalling pathway—mediates the effects of L-arginine on ISC function. To test the specificity of L-arginine's effects in our study, examining the effect of other amino acids is necessary. Moreover, previous studies have shown that glutamine supplementation could regulate the function of ISCs in mice[58,59]. CD90+ stromal cell media was supplemented with 1 mM of different amino acids (L-arginine, L-leucine, L-lysine, or L-glutamine). Interestingly, both L-arginine and L-leucine up-regulated expression of WNT2B protein. However, L-lysine and L-glutamine had no effects. These results indicated that the effects of L-arginine on CD90+ stromal cells are specific to some degree. However, future studies are needed to further explore the specificity of L-arginine's effects on ISCs.

In mammals, L-arginine is particularly important, promoting diverse physiological effects such as protein synthesis that are largely mediated through activation of mTOR[36]. mTOR is a sensor of nutrients and growth factors. mTORC1 activation promotes cell proliferation by increasing global protein synthesis and other anabolic processes[36,60]. mTORC1 signalling is required for intestinal epithelial cell proliferation during homeostasis and regeneration[61–64], including regeneration mediated by quiescent ISCs[65,66]. In addition, several studies have shown that caloric restriction (CR) promotes Lgr5+ ISC expansion via mTORC1 signalling. In the intestine, CR was reported to increase the number of ISCs by reducing mTORC1 signalling in Paneth or niche cells[67]. Unexpectedly, another publication reported that mTORC1 activity in ISCs is up-regulated during CR, resulting in increased protein synthesis in ISCs and an increase in their number[68]. Down-regulation of mTORC1 in Paneth cells is necessary for ISC pool expansion, whereas up-regulation of mTORC1 in ISCs is necessary to respond and drive the increase in ISC number. These studies indicated that the regulation of mTORC1 in different cell types is truly complex in the ISC niche and may depend on the dose and type of mTOR agonist or inhibitor, the age and nutritional status of animals and so on. For example, we found that low dose L-arginine supplementation (1 mM) stimulated CD90+ stromal cells to secrete

Wnt2b and promote Lgr5+ ISC-mediated epithelium renewal. However, high doses of L-arginine supplementation (2 mM, 3 mM) destroyed intestinal organoids. Consistent with other reports, we found that L-arginine supplementation activated the mTOR signalling pathway in CD90+ stromal cells. Moreover, expression levels of Wnt2b in stromal cells was highly activated after addition of L-arginine and the mTOR activator MHY1485 in vitro. In addition, L-arginine and Wnt2b protected SI organoids from epithelial damage provoked by TNF-α. This phenomenon further confirmed our hypothesis that L-arginine stimulates proliferation of ISCs to maintain integrity of the intestinal epithelial barrier in a Wnt2b-dependent manner.

5-FU has been used to treat many kinds of cancers and can result in damage to the SI epithelium. Dietary L-arginine supplementation before 5-FU treatment protected mice against the destruction of villi and crypts. It has been reported that 5-FU induces marked apoptosis in Lgr5+ stem cells in mice[69]. In this study, we also observed that most adult Lgr5+ ISCs had disappeared by 5 days after 5-FU treatment. However, pre-treatment with L-arginine enhanced survival of Lgr5+ ISCs cells in vivo. Meanwhile, crypts from 5-FU-injected mice pre-treated with L-arginine were more likely to form organoid bodies. L-arginine supports Lgr5+ ISC survival and subsequently results in efficient protection in response to gut injury.

In summary, we found that regulation of Wnt2b in the CD90+ stromal cell niche by exogenous L-arginine mediates ISC function. Our results also established a model in which the mammalian ISC niche couples organismal nutrient levels to stem cell function. By taking advantage of the essential roles of the L-arginine–ISC niche, we propose clinical use of L-arginine to protect against gut injury in response to stimuli, such as chemotherapy. Although the importance of Wnt2b/mTOR signals were observed, our conclusions still need further validation using Wnt2b-knockout animals. Moreover, future studies will need to further explore (1) the metabolic pathways of L-arginine in stromal cells, (2) the roles of exogenous L-arginine on immune, and (3) microbial elements in the stem cell microenvironment.

## Methods
**Mice**. Male C57BL/6N (specific-pathogen-free [SPF]) mice (8–12 weeks old) were purchased from the Laboratory Animal Centre of the Institute of Genetics (Beijing, China). Male Lgr5-EGFP-IRES-CreERT2 (Lgr5-GFP) mice (4–8 weeks old) (specific-pathogen-free [SPF]) were purchased from Jackson Laboratory. Mice were maintained according to the Guide for the Care and Use of Laboratory Animals (Institute for Learning and Animal Research at China Agricultural University; SYXK-2015-0028). All procedures were performed in accordance with institutional and national guidelines and regulations and were approved by the China Agricultural University Animal Care and Use Committee.

**Crypt isolation and cell dissociation**. Isolation of intestinal crypts and dissociation of cells for flow cytometry analysis were largely performed as previously described[70]. In brief, after euthanizing mice with $CO_2$ and collecting the small

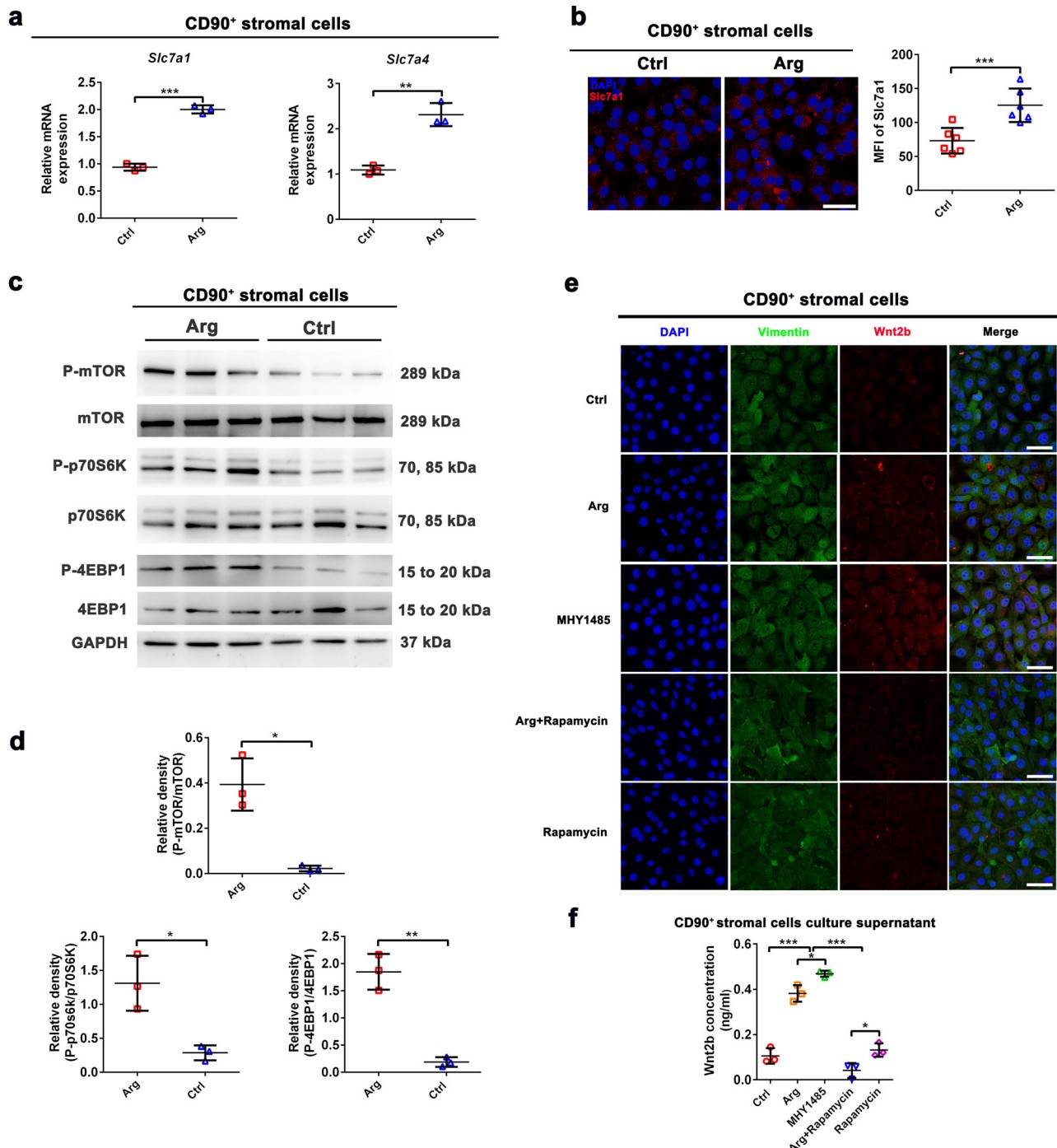

**Fig. 6 L-arginine supplementation stimulates Wnt2b synthesis in CD90$^+$ stromal cells through the mTORC1 signalling pathway.** CD90$^+$ stromal cells were cultured in medium supplemented with L-arginine (1 mM) for 3 days. **a** qPCR of relative mRNA expression of *Slc7a1* and *Slc7a4* genes in CD90$^+$ stromal cells treated with L-arginine or not. Expression show is relative to *GAPDH* gene, $n = 3$. **b** Immunostaining of Slc7a1 (red) and DAPI (blue) in CD90$^+$ stromal cells treated with L-arginine or not. Scale bar, 10 μm. Quantitation for Slc7a1 was measured by MFI, $n = 6$. **c**, **d** Protein levels of GAPDH, mTOR, P-mTOR, P70S6K, P-P70S6K, 4EBP1, and P-4EBP1 were measured by western blotting assay in CD90$^+$ stromal cells treated with L-arginine or not, $n = 3$. **e** Immunostaining of Wnt2b (red), Vimentin (green), and DAPI (blue) in CD90$^+$ stromal cells treated with L-arginine (1 mM), MHY1485 (5 μM), Rapamycin (5 μM) and their combinations for 3 days. Scale bar, 10 μm. **f** The concentration of Wnt2b in CD90$^+$ stromal cells was detected by ELISA, $n = 3$. Data are the mean ± SD; comparisons performed with *t*-tests (two groups) or analysis of variance (ANOVA) (multiple groups). *$P < 0.05$, **$P < 0.01$, ***$P < 0.001$. Results are representative of two or three independent experiments.

intestines, organs were opened longitudinally and washed with PBS. To dissociate the crypts, the small intestine was incubated at 4 °C in EDTA (0.5 mM) for 30 min. To isolate single cells from the small intestinal crypts, the pellet was further incubated in 1× TrypLE express (Gibco, Life Technologies) supplemented with 0.8 kU ml$^{-1}$ DNase1 (Roche).

**Murine intestinal organoid isolation and culture**. According to methods described previously, we isolated fresh SI crypts from 8-week-old male C57BL/6N mice[46]. For mouse organoids, depending on the experiments, 200–400 crypts per well were suspended in 50 μl Matrigel composed of 50% advanced DMEM/F12 medium (Gibco) and 50% growth-factor-reduced Matrigel (Corning). The Matrigel

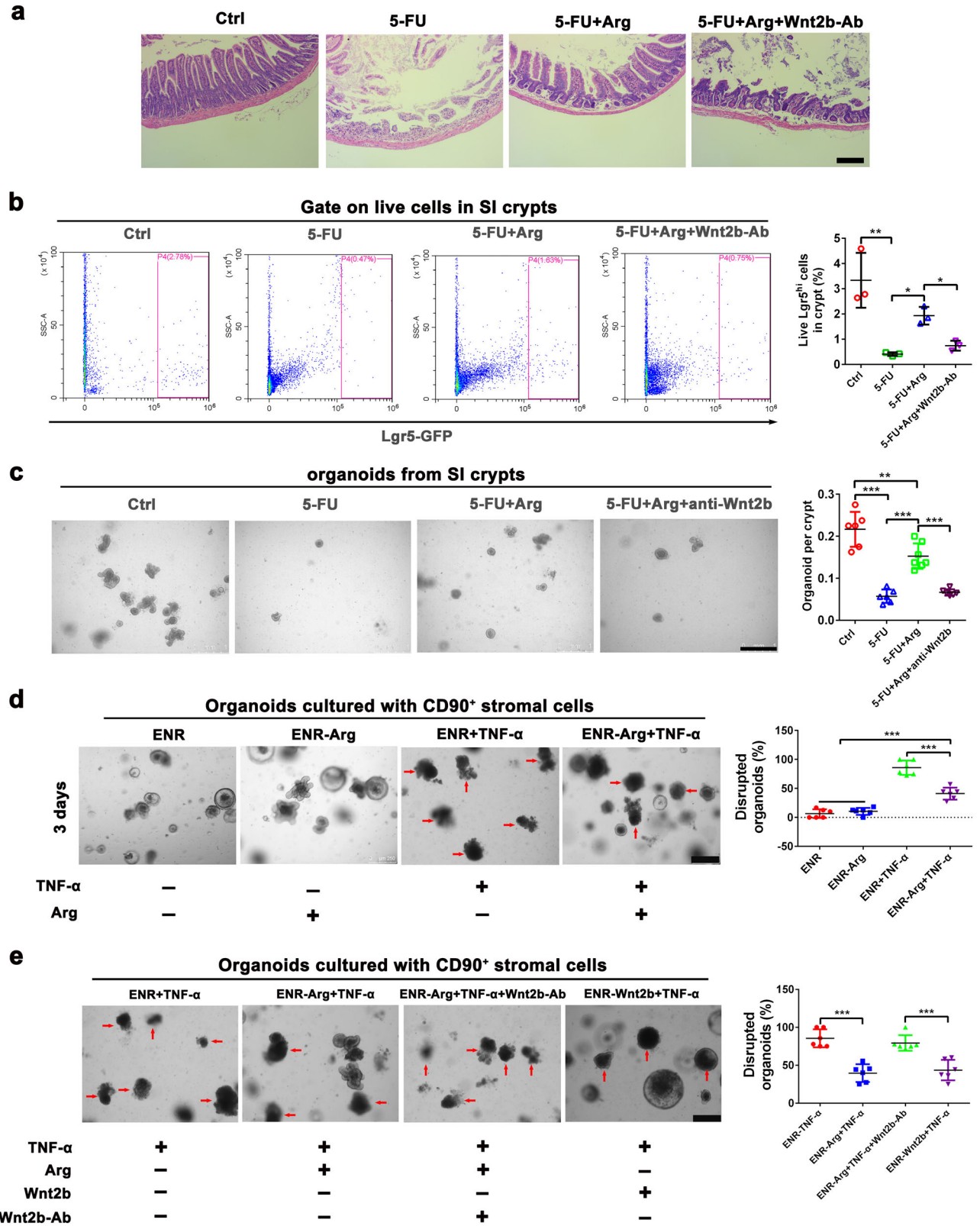

suspension was allowed to polymerize at 37 °C for 30 min before adding 500 μl fresh complete medium (advanced DMEM/F12 (Gibco) supplemented with 100 U/ml penicillin-100 g/ml streptomycin (Gibco), 10 mM HEPES (Gibco), N-2 (Gibco), B-27 (Gibco), 50 ng/mL mouse EGF (Peprotech), 100 ng/ml mouse Noggin (Peprotech) and 500 ng/ml mouse R-spondin1 (Peprotech)). Culture medium and growth factors were replaced every 3–4 days. Clonogenicity (colony-forming efficiency) was calculated by plating 200–400 crypts and assessing organoid formation after 3–7 days or as specified after initiation of cultures.

Lgr5–GFPhi stem cells from 8-week-old male Lgr5-EGFP-ires-creERT2 (Lgr5-GFP) mice[42] were sorted. In brief, single cells were isolated from Lgr5–GFP mice using a modified crypt isolation protocol with 5 mM EDTA for 20 min followed by several strainer steps and a 5-min incubation with TrypLE and 0.8 kU ml$^{-1}$ DNase1 under minute-to-minute vortexing to create a single-cell suspension. The cell suspensions were stained with 7-AAD (Biolegends) for flow cytometry analysis to exclude dead cells. Gating for 7-AAD$^{-}$ cells was used to determine and sort Lgr5hi cells. GFP$^{+}$ analysis gates were established so that Lgr5-GFPhi events were

**Fig. 7 L-arginine supplementation protects the gut from 5-FU and TNF-α induced intestinal epithelial damage in a Wnt2b-dependent manner. a–c** Mice were treated with normal water or 7 mg/ml L-arginine in their drinking water for 14 days and then injected with 5-FU (300 mg/kg) or PBS as a control once a day for 5 days. In Wnt2b antibody neutralization experiment, every mouse were given 300 μg Wnt2b neutralizing antibody once on the first day of 5-FU injection. **a** Pathology of the small intestine by H&E staining. Scale bar, 200 μm. **b** FCM analysis of live Lgr5hi ISC frequency in SI crypts, n = 3. **c** SI organoid frequency of crypts from mice in different groups. Representative images of crypt culture from each group are shown, n = 6. Scale bar, 50 μm. **d–e** SI organoids-CD90+ stromal cells co-culture model was cultured with ENR-medium or ENR-medium supplemented L-arginine (1 mM), TNF-α (100 ng/ml), WNT2B protein (100 ng/ml), Wnt2b neutralizing antibody (1 μg/ml) and their combinations for 3 days. **d** The light microscope observation of the co-culture model treated with L-arginine, TNF-α, and their combinations. Scale bar, 200 μm. The number of total organoids and disrupted organoids with altered morphology per well were counted, n = 6. **e** The light microscope observation of co-culture model treated with L-arginine, TNF-α, WNT2B protein, Wnt2b neutralizing antibody and their combinations. Scale bar, 200 μm. The number of total organoids and disrupted organoids with altered morphology per well were counted, n = 6. Data are the mean ± SD; comparisons performed with t-tests (two groups) or analysis of variance (ANOVA) (multiple groups). *P < 0.05, **P < 0.01, ***P < 0.001. Results are representative of two or three independent experiments.

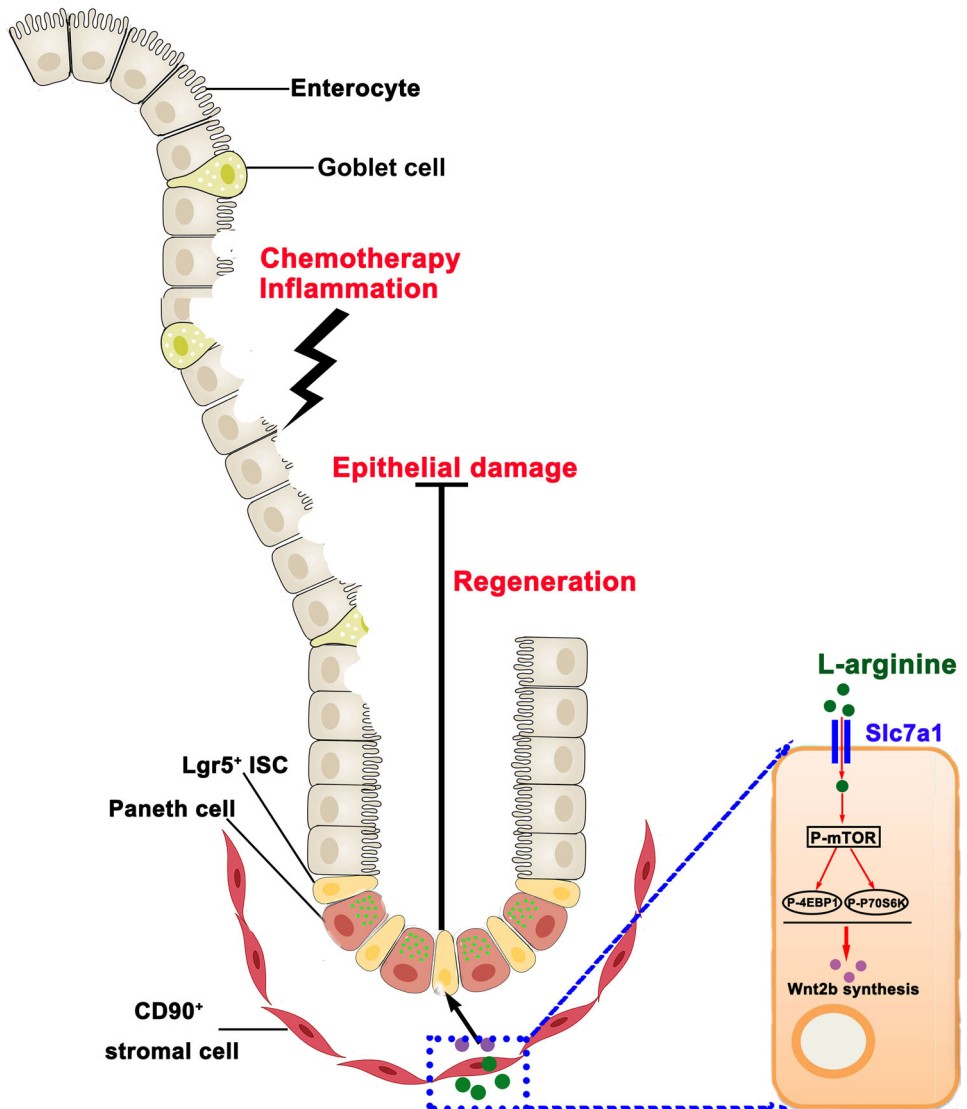

**Fig. 8 L-arginine increased expansion of ISCs in SI crypts and accelerates ISC-mediated intestinal epithelial regeneration.** By constructing a co-culture model, we found that CD90+ stromal cells, a key constituent of the mammalian ISC niches, augmented stem-cell function in response to L-arginine. Moreover, increased expression of Wnt2b in CD90+ stromal cells mediated the effects of L-arginine on ISC function. We found that L-arginine stimulated Wnt2b secretion of CD90+ stromal cells through mTORC1 signals.

detectable in intestinal crypts and organoids from Lgr5-GFP mice. Isolated ISCs were centrifuged at 250 × g for 5 min, re-suspended in the appropriate volume of crypt culture medium and seeded onto 50 μl Matrigel (Corning) containing 1 mM JAG-1 protein (R&D system) in 24-well plate. Crypt media was changed every second or third day. Organoids were quantified on day 3 unless otherwise specified.

**Murine intestinal stromal cell isolation and culture**. The small intestine from 8-week-old male C57BL/6N mice was collected from the duodenum to the cecum, cut open length-wise and washed in cold PBS. After incubation with 35 ml 0.5 mM EDTA in DPBS for 30 min at 4 °C, tissues were subjected to an additional round of pipetting (40 times) to remove most of the remaining epithelial cells and washed

twice with PBS. Thereafter, tissues were digested in 10 ml DMEM complete media (5% FBS, penicillin/streptomycin, HEPES, L-glutamine) containing 2 mg/ml of Collagenase/Dispase (Roche) at 37 °C for 1 h. The digestion solution was replaced with fresh digestion solution every 30 min. At the end of this digestion step, tissues were further dissociated by vigorous pipetting. Thereafter, samples were passed through a 70 μm cell strainer, centrifuged at $400 \times g$ for 4 min, washed once in PBS and counted. Percoll gradient centrifugation was performed to remove non-cellular debris. Cell pellets were resuspended in RPMI with 30% Percoll (Solarbio), and the resulting suspension was layered over PBS with 70% Percoll (Solarbio) in a 15 mL Falcon tube. CD45 (Miltenyi, 130-052-301) conjugated MicroBeads were added to the cell suspension to exclude intestinal leukocytes, and the mixture was incubated at 4 °C for 30 min. Cells were then washed and loaded onto an LD column via a 35 mm pre-separation filter. The flow-through fraction was collected and then incubated with CD90 conjugated MicroBeads at 4 °C for 30 min. Cells were then washed and loaded onto an LD column via a 35 mm pre-separation filter. The non-flow-through was collected, centrifuged at $400 \times g$ for 4 min, and washed once in PBS. CD90$^+$ stromal cells were cultured for 5 days in RPMI1640 containing 10% FCS, 1% penicillin-streptomycin and 1% Glutamax. For organoid co-cultures, stromal cells were plated in 24-well plates for 24 h before ISC isolation, and ISCs resuspended in Matrigel were added on top of the cell layer (5000 stromal cells and 100 ISCs per well). To detect the effect of L-arginine on ISC function, SI organoids and the ISC-stromal cell co-culture model were cultured with ENR-medium or ENR-medium supplemented with 1 mM L-arginine. The growth status and morphology of SI organoids were observed under light microscopy.

**Quantitative RT-PCR**. SI organoids and crypts were harvested after treatment. Total RNA was extracted from the SI organoids and crypts using RNAiso Plus (Takara). Reverse transcription of RNA was performed with the primers listed in Supplementary Table 1. Two microliters of template RNA were reacted with SYBR PCR Master Mix in a final volume of 20 μl (Takara). The thermal cycling conditions were 5 min at 95 °C, followed by 40 cycles of 15 s at 95 °C and 34 s at 60 °C using an Applied Biosystems 7500 real-time PCR system.

**Western blot**. Different treatment of SI organoids were lysed in RIPA buffer (50 mM Tris-HCl (pH 7.4), 1% NP-40, and 150 mM NaCl) containing a protease inhibitor cocktail (Thermo Scientific). Protein concentrations were detected using a BCA protein quantification kit (Thermo Scientific). Equal amounts of protein were separated by SDS-PAGE and electrophoretically transferred onto PVDF membranes (Millipore). After blocking with 5% non-fat milk in TBS containing 0.1% Tween-20, membranes were probed with rabbit anti-non-phospho (Active) β-Catenin (Ser45) (CST, 1:1000; 19807), rabbit anti-mTOR (CST, 1:1000; 2983), rabbit anti-phospho-mTOR (CST, 1:1000; 2974), rabbit-anti-p70S6 kinase (CST, 1:1000; 9202), mouse anti-phospho-p70S6 kinase (CST, 1:1000; 9206), rabbit anti-4EBP1 (CST, 1:1000; 9644), rabbit anti-phospho-4EBP1 (CST, 1:1000; 2855) and mouse anti-GAPDH (Solarbio, 1:1000; K200057M). After washing, membranes were incubated with goat anti-rabbit secondary antibodies (Beyotime, 1:5000; A0208) and goat anti–mouse secondary antibodies (Beyotime, 1:5000; A0216). Signals were detected using a SuperSignal West Pico kit (Thermo Scientific) and subjected to an Image Reader LAS-4000 imaging system (FUJIFLIM).

**Cytokines and L-arginine detection**. LPS and TNF-α levels in serum or intestine were measured using ELISA kits (eBioscience) according to the manufacturer's instructions. WNT2B levels were measured using an ELISA kit (CUSABIO) according to the manufacturer's instructions. L-arginine concentrations in serum or intestine were measured using an L-Arginine Assay Kit (Abcam, ab241028) according to the manufacturer's instructions.

**Mouse experiments**. Lgr5-GFP mice (8-week old) were fed drinking water containing 7 mg/ml L-arginine for 2 weeks. Then, mice were sacrificed, and subsequent experiments were performed. Detailed methods are shown in Fig. 1a. In brief, mice were sacrificed and their small intestines were removed with the SI length and mass recorded. Tissues were fixed in 4% paraformaldehyde, embedded in paraffin wax, and sliced. SI crypts were isolated from mice, and clonogenicity was calculated by plating 200–400 crypts and assessing organoid formation after 3–5 days or as specified after initiation of cultures.

Lgr5-GFP mice (8-week old) were fed drinking water containing 7 mg/ml L-arginine for 2 weeks, and then mice were injected with 5-FU (300 mg/kg, Sigma) once a day for 5 days. Then mice were sacrificed, and subsequent experiments were performed. The detailed methods are listed in Fig. 6a. In brief, mice were sacrificed, and then their small intestine was removed. Jejunal tissues were fixed in 4% paraformaldehyde, embedded in paraffin wax, sliced, and stained with HE. SI crypts were isolated from mice, and clonogenicity was calculated by plating 200–400 crypts and assessing organoid formation after 3–7 days or as specified after initiation of cultures.

**Immunofluorescence assay**. SI organoids were rinsed three times in ice-cold PBS and then suspended in cold PBS. SI organoids were spun down at 900 rpm for 10 min at 4 °C and then fixed overnight in 4% paraformaldehyde. Sections of the small intestine (2 cm each) were collected from mice in different groups, fixed overnight

in 4% paraformaldehyde, OCT (optimal cutting temperature compound)-embedded, sectioned at 8 μm-thick and rinsed in PBS. Intestinal organoids and tissue sections were permeabilized in 0.5% Triton X-100 for 20 min, washed three times in PBS and incubated for 1 h in 3% BSA in PBS to reduce nonspecific background signals. For IF staining, SI organoids, stromal cells, and tissue sections were incubated overnight with primary antibodies (anti-Olfm4, 1:400, CST, 39141; anti-lysozyme, 1:400, Abcam, ab108508; anti-Muc2, 1:400, Abcam, ab272692; anti-Vimentin, 1:100, CST, 5741; anti-Wnt2b, 1:50, Abcam ab246361; Anti-CD90, 1:100, Abcam, ab225; anti-non-phospho (Active) β-catenin (Ser45), 1:400, CST, 19807; and anti-Slc7a1, 1:50, Abcam, ab37588). Samples were then incubated with Alexa-488, −594 and −647–conjugated anti-rabbit IgG and anti-mouse IgG antibodies (Beyotime, 1:500) for 1 h followed by DAPI (1:1000, Solarbio, Hoechst33342) for 5 min at room temperature. Samples were examined using a Zeiss 710 or Leica TCS SP8 laser-scanning confocal microscope. Fluorescence images were collected for further qualitative and quantitative analyses. Quantification of mean fluorescence intensity (MFI) was performed by ImageJ. Numbers of Lgr5$^+$, Olfm4$^+$, lysozyme$^+$, Muc2$^+$ and EdU$^+$ cells per bud and crypt were manually counted.

**Statistics and reproducibility**. Results are presented as means ± SD. One-way ANOVA was employed to determine statistical differences among multiple groups, and $t$-test was employed to determine differences between the two groups. *$P <$ 0.05, **$P < 0.01$, ***$P < 0.001$. Data are representative of two or three independent experiments unless otherwise stated. The sample size is indicated for each experiment in the corresponding figure legend.

**Reporting summary**. Further information on research design is available in the Nature Research Reporting Summary linked to this article.

## Data availability
Data supporting the findings of this work are available within the paper and its Supplementary files or are available from the corresponding author upon reasonable request. Raw data underlying plots in the figures are available in Supplementary Data 1. Unprocessed blot images with size markers are provided in Supplementary Information.

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

## Acknowledgements

This work was supported by National Key R&D Program of China (2017YFE0129900) and the Funding of Young Talent Supporting Program of the College of Animal Science and Technology of the China Agricultural University Education Foundation (2017DKA002).

## Author contributions

Q.H. was responsible for the design of the study, performing the experiments, data analysis, and writing the manuscript; Y.D., J.H., L.Y., Y.W., Y.H., J.L., J.S., M.S., and Q. J. participated in the animal experiments and data collection; B.W., J.H., Y.B., and Y.L. helped write the manuscript; Z.Y. and Y.G. helped design experiments; B.Z. was responsible for the conception and design of the study, drafting the article, and final approval of the version submitted. All authors have read, commented on, and approved the final paper.

## Competing interests

The authors declare no competing interests.
