## [Peer Review File · Communications Biology]

Reviewers' comments:

Reviewer #1 (Remarks to the Author):

In this manuscript, Hou et al. show that L-Arginine regulates intestinal stem cell function through Wnt2b secretion by CD90+ intestinal stromal cells. They further go on to show that L-Arginine has protective effects on intestinal epithelium in response to both TNF-alpha and 5-Fluorouracil induced damage. Finally, they propose a mechanism through which L-Arginine dependent mTORC1 activation promotes Wnt2b synthesis in the stromal cells. Although the observations that L-Arginine protects intestinal mucosa from damage have been previously shown in several studies, the authors make important and further mechanistic insights with this study that may have potential therapeutic applications. However, several key experiments and conclusions need to be further completed to solidify these conclusions.

Major Comments:

1. Could the authors explain the methods for measuring and normalizing L-arginine in serum and tissues for Figure 1B? For the y-axis units - do they mean uM?
2. Can the authors re-assess the numbers of Paneth and Goblet cell numbers with more cell specific markers and staining? How do they differentiate between PAS+ secretory subtypes?
3. It is unclear from the methods how the authors set up the Flow cytometry analysis for LGR5 hi stem cells. Do they mean they sorted the cells based on GFP or using an Lgr5 antibody (Figure 1E)? Can they elaborate in the methods?
4. From figure 1D, it is hard to tell if there are more proliferating stem and/or progenitor cells. Can the authors include insets for these images that focus on the crypts at higher magnification and resolution?
5. Conversely, can the authors also please include in the supplemental data full tissue images at lower magnification representative of the crypts used for Figure 1F?
6. Figure 2F is not discussed in the manuscript. In addition, where is figure 2I and J that the authors refer to?
7. In Figure 2 the authors test whether exogenous L-Arginine treatment directly augments intestinal stem cells. The authors need to rule out or in whether there are any effects of L-Arginine treatment mediated through Paneth cells, which is not tested here, as they only test the effects on intestinal stem cells directly. In addition, can the authors show the effects of in vivo L-Arginine treatment on stem cell clonogenicity?
8. In Figure 3D it looks as if the ISC are in close contact with the stromal cells, however, in 3E it is seems that in these crypt cultures the stromal cells are not seen as in Fig 3E. Can the authors explain the differences between the two culture methods?
9. Similar to Figure 2, can the authors sort and mix stromal cells from the L-Arg treated animal with non-treated stem cells to further solidify conclusion that effects are only coming from the stromal population?
10. In figure 4E, it looks as though IWP-2 alone suppresses organoid formation independent of Arg effects. Can the authors comment on that? Also, based on Fig S2B there may still be some Arg dependent partial effects on Wnt dependent genes.
11. "However, L-arginine treatment alleviated and protected damage induced by TNF- α on SI

organoids to maintain normal morphology". The protective effects of Arg appear to be somewhat partial here, so the authors should change the conclusions accordingly.

12. Can the authors elaborate and discuss how their findings fit with previous studies that show pharmacological and dietary mTORC1 suppression has both direct and indirect effects on intestinal stem cells (Yilmaz et al, Igarashi et al). This also goes back to the question of how L-arginine affects Paneth cell responses.

Minor Comments:

1. In the abstract the authors state that L-Arginine may have a potential as a therapeutic for intestinal regeneration without stating why that may be. Could the authors add a transition sentence as why they hypothesized L-arginine may have a potential as a therapy for mucosal injury?
2. Fig S1A is missing units for the x axis.
3. There are some typos throughout the manuscript that should be fixed.
4. Some of the figures have omitted axis labels and wrong units.

Reviewer #2 (Remarks to the Author):

In this manuscript, Hou et al. investigate the effect of L-arginine on the Lgr5-positive intestinal epithelial stem cell. Previous studies had indicated that L-arginine supplementation was protective against 5-FU injury, but whether this was due to an effect on ISC function and the mechanism of the protection were unclear. The authors found that oral L-arginine supplementation in mice resulted in increased numbers of Lgr5+ stem cells and proliferating Lgr5+ stem cells. This effect was only observed using in vitro organoid culture if CD90+ stromal cells were also present in the culture and was linked to the secretion of Wnt2b from these critical niche cells. The authors then provide data that the altered Wnt2b secretion from the CD90+ stromal cells was due to activated mTOR signaling. Finally, the authors added to existing literature on the protective effect of L-arginine in the 5-FU injury model by demonstrating that increased ISC activity is maintained in L-arginine pre-treated mice. This is overall a very nice study that will be of interest to the community. However, there are areas where the methods or experimental details are unclear, which need to be clarified in order to support the authors' interpretation of these data.

Major comments:

What is the rationale for the chosen dose of 7 mg/mL of L-arginine? Also, the control treatment is not stated. Were the control mice given normal drinking water alone? If so, could more specific controls be tested in mice and/or in cultured cells to give some indication of the specificity of this response to the L-arginine vs. other amino acids?

How were L-arginine concentrations determined in Fig. 1B? I can't find this information in the manuscript.

In the images in Fig. 1D, is the green line supposed to indicate what was counted as "crypt"? It appears that the tops of many crypts were not included in the counted region. How was the counted region defined?

Fig 1E: Please include the full gating strategy for the identification of the Lgr5+ cells by flow cytometry. It is not clear how the current gate was drawn, as it appears to be in the middle of the Lgr5 positive-expressing population in the images shown. Also, please follow the journal Reporting Guidelines for flow cytometry – add Lgr5-GFP to the axis label, etc.

The Lgr5-GFP mouse is known to express GFP in a heterogeneous fashion. How was this handled experimentally to generate the data in Fig. 1F? How many crypts were counted? What was done if a crypt had no GFP-expressing cells?

The conclusions of Fig 2 could be further supported by including data on the Paneth cells present in organoids +/- L-arginine treatment, as this would be the other key epithelial cell type known to support stem cell function by producing Wnts. Also, did the authors observe any change in mRNA markers for either the Lgr5 stem cell or +4 quiescent intestinal epithelial stem cell?

Fig 4A,B: If Lgr5-GFP would be on the green channel and b-catenin is also on the green channel, then how was only b-catenin expression quantified in this experiment? Perhaps I am missing something.

Fig 4C: As nuclear b-catenin is a key proliferative signal, these Western blots should be redone using nuclear preps.

Fig 4F: This b-catenin IF seems abnormal. Why is the plasma membrane not staining? It appears that an antibody that generally marks all b-catenin was used, and thus basolateral staining would be expected.

Can the authors provide any data to suggest that Wnt2b/mTOR activation in CD90+ stromal cells are required for the protective effect of L-arginine in the 5-FU in vivo experiment?

Minor comments:

Please indicate the sex of mice used in these experiments.

Were tests performed to test for parametric vs. non-parametric data distributions to ensure the validity of the 1-way ANOVA and t tests performed in this study?

Fig 1G: Please mark the figures to indicate what was counted as a live organoid, as there appears to also be dead/non-organoid material in these images.

In the Reporting Guidelines form, the authors state that a minimum of 3 experiments were performed, but in the manuscript text, they state that a minimum of 2 experiments were performed. Which is it? Please show individual values in graphical data rather than using bar graphs to make this more clear.

Manuscript: COMMSBIO-20-0325

Title: Regulation of mTORC1 pathway in CD90⁺ stromal cells by exogenous L-arginine couples ISC function through Wnt2b/ β -catenin pathway

Communications Biology

Thanks very much for your and reviews' kind suggestion, which are helpful for us to improve our manuscript. We have performed extra experiments according to reviewers' suggestions. The point to point responses are listed below.

Reviewers' comments:

Reviewer #1 (Remarks to the Author):

In this manuscript, Hou et al. show that L-Arginine regulates intestinal stem cell function through Wnt2b secretion by CD90⁺ intestinal stromal cells. They further go on to show that L-Arginine has protective effects on intestinal epithelium in response to both TNF-alpha and 5-Fluorouracil induced damage. Finally, they propose a mechanism through which L-Arginine dependent mTORC1 activation promotes Wnt2b synthesis in the stromal cells. Although the observations that L-Arginine protects intestinal mucosa from damage have been previously shown in several studies, the authors make important and further mechanistic insights with this study that may have potential therapeutic applications. However, several key experiments and conclusions need to be further completed to solidify these conclusions.

A: Thanks very much for your careful revision. According to your suggestions, we performed experiments to improve the quality of our manuscript. Please find the following detailed responses to your comments and suggestions.

Major Comments:

1. Could the authors explain the methods for measuring and normalizing L-arginine in serum and tissues for Figure 1B? For the y-axis units - do they mean μ M?

A: We are sorry for missing this important information. Thanks very much for your

reminding. We utilized the L-Arginine Assay Kit (Abcam, ab241028) to measure total L-arginine concentrations in various samples according to the manufacturer's instructions. We have added "L-arginine measurement" in the **MATERIALS AND METHODS** section. The y-axis units in Figure 2B indeed mean " μM ". We have corrected this mistake. Thanks! Protocol summary is as below.

1. Prepare small intestinal content, small intestinal tissue, and serum samples, controls and standards as instructed.
2. Prepare the Arginine standard curve.
3. Add 20 μL of sample to desired wells, adjust volume to 40 μL with Arginine Assay Buffer.
4. Prepare the Enzyme mix and add 10 μL to each well containing Standard, Sample, Sample Background and Spiked wells. Prepare the Reaction Mix.
5. Add 200 μL to Standard, Sample, Sample Background and Spiked wells.
6. Measure the absorbance (450 nm) in a microplate in end point mode.

2. Can the authors re-assess the numbers of Paneth and Goblet cell numbers with more cell specific markers and staining? How do they differentiate between PAS+ secretory subtypes?

A: Thanks for your constructive suggestion, which is valuable for improving the accuracy of the manuscript. We have re-assessed the numbers of Paneth and goblet cell with more specific markers. We stained Paneth cells and goblet cells with Lysozyme antibody (Abcam, ab108508) and Muc2 antibody (Abcam, ab272692), respectively. Consistent with PAS staining results, L-arginine supplementation caused a commensurate increase in Lysozyme⁺ Paneth cells in small intestinal crypt (**Fig. S1C**), but mildly reduced that of Muc2⁺ goblet cells (**Fig. 1D**). Thanks!

3. It is unclear from the methods how the authors set up the Flow cytometry analysis for LGR5 hi stem cells. Do they mean they sorted the cells based on GFP or using an Lgr5 antibody (Figure 1E)? Can they elaborate in the methods?

A: Lgr5-GFP^{hi} stem cells were sorted from Lgr5-EGFP-ires-creERT2 (Lgr5-GFP)

mice¹. In briefly, single cells were isolated from Lgr5-GFP mice using a modified crypt isolation protocol with 20 min of 5 mM EDTA followed by several strainer steps and a 5-min incubation with TrypLE and 0.8 kU ml⁻¹ DNase1 under minute-to-minute vortexing to make a single-cell suspension. The cell suspensions were stained with 7-AAD (7-amino-actinomycin D) (Biolegends) for flow cytometry analysis to exclude dead cells. Gating for 7-AAD⁻ cells was used to determine and sort Lgr5^{hi} cells. GFP⁺ analysis gates were established so that Lgr5-GFP^{hi} events were detectable in intestinal crypts and organoids from Lgr5-GFP mice. Meanwhile, we have elaborated this in the methods. Thanks for your useful suggestion.

4. From figure 1D, it is hard to tell if there are more proliferating stem and/or progenitor cells. Can the authors include insets for these images that focus on the crypts at higher magnification and resolution?

A: In figure 1D, we circled the crypt area with white dotted lines. A higher percentage of 5-Ethynyl-2'-deoxyuridine (EdU)-positive cells was observed in small intestinal crypts from mice fed L-arginine. This data only indicated there were more proliferating intestinal epithelium cells in the crypt area. Overall, data in Figure 1 demonstrated that L-arginine supplementation could promote ISC-mediated intestinal epithelial regeneration. Thanks for your kindly suggestion.

5. Conversely, can the authors also please include in the supplemental data full tissue images at lower magnification representative of the crypts used for Figure 1F?

A: Thanks for your kind suggestion. We noticed that the use of Lgr5-GFP mice to count Lgr5⁺ stem cells in intestinal crypts must take into account the mosaicism of GFP expression in the intestines of these mice, which varies from mouse to mouse. If a crypt is completely negative, this may simply be due to its mosaic expression, so only using Lgr5⁺ cell for quantitation is not very accurate. Olfactomedin-4 (Olfm4) can serve as a useful marker for Lgr5-type stem cells in small intestine and colon². Hence, we used OLFM4 antibody (CST, 39141) to stain mouse intestinal stem cells (**Fig. 1F**). We obtained the small intestinal tissues from 3 mice per group and every

20-30 crypts were counted per tissue. We have added both higher and lower magnification of the crypts in Figure 1F. This result demonstrated that L-arginine supplementation significantly increased numbers of Olfm4⁺ ISCs in small intestinal crypts when compared to control group. Thanks!

6. Figure 2F is not discussed in the manuscript. In addition, where is figure 2I and J that the authors refer to?

A: I am sorry for the mistake. Thanks very much for your carefully revision. We have carefully checked all the Figures in the manuscript and made corrections for errors. Thanks!

7. In Figure 2 the authors test whether exogenous L-Arginine treatment directly augments intestinal stem cells. The authors need to rule out or in whether there are any effects of L-Arginine treatment mediated through Paneth cells, which is not tested here, as they only test the effects on intestinal stem cells directly. In addition, can the authors show the effects of in vivo L-Arginine treatment on stem cell clonogenicity?

A: Thanks for your constructive suggestion, which is valuable for improving the accuracy of the manuscript. Previous studies demonstrates that Paneth cells provide important factors for intestinal stem cell (ISC) maintenance³, such as DLL1/4, TGF- α , Wnt3^{4,5,6}. The enhanced regenerative activity of ISCs in mice fed L-arginine led us to ask whether ISCs responded to L-arginine through the Paneth cell niches. However, exogenous L-Arginine (1 mM) treatment had no effect on numbers of Lysozyme⁺ Paneth cell in organoids (**Fig. S2C**). Similar results were further verified by the mRNA expression of Paneth cell marker genes (*Lyz1*, *Defa6*) (**Fig. S2D**). Moreover, we found that L-arginine supplementation have no significant impact on the mRNA expression of *Wnt2b*, *Wnt3*, *Axin2*, and *Ctnnb1* (**Fig. S2E**). Collectively, these results indicated that the effects of exogenous L-arginine treatment on ISC

function might not be mediated through Paneth cells niche. We also tested the potential of sorted Lgr5⁺ ISCs from Lgr5-GFP mice to form clonal organoid bodies. Sorted Lgr5⁺ ISCs from Lgr5-GFP mice treated with L-arginine were more likely to form organoid bodies than those from the control group (**Fig. 1H**). Thanks!

8. In Figure 3D it looks as if the ISC are in close contact with the stromal cells, however, in 3E it is seems that in these crypt cultures the stromal cells are not seen as in Fig 3E. Can the authors explain the differences between the two culture methods?

A: Thanks for your kind question. As shown in Figure 3A and 3C, CD90⁺ stromal cells were plated on 24-well plate for 24 h before Lgr5⁺ ISC isolation, and Lgr5⁺ ISCs resuspended in Matrigel were added on top of the cell layer (5000 stromal cells and 100 ISCs per well). The growth status and morphology of SI organoids were observed under the Fluorescence Microscope. This co-culture model is a three-dimensional culture system. However, the Fluorescence Microscope can only see a two-dimensional visual field. Therefore, as shown in Figure 2C, when ISCs and stromal cells were just mixed, we could see little ISCs are in close contact with the stromal cells in one visual field. However, when the ISCs develop into three-dimensional organoids, if we want to observe the organoids, we can not see the stromal cells, such as pictures in Figure 2D, 2F and 2J. Because they are not in one two-dimensional visual field. We sincerely hope that our explanation has addressed your question. Thanks!

9. Similar to Figure 2, can the authors sort and mix stromal cells from the L-Arg treated animal with non-treated stem cells to further solidify conclusion that effects are only coming from the stromal population?

A: Thanks for your constructive suggestion. To further solidify our conclusion that the effects of exogenous L-arginine on ISCs are only coming from the stromal population, we sorted and mixed CD90⁺ stromal cells from the L-arginine treated mice with non-treated Lgr5⁺ ISCs and assayed their ability to form organoid bodies *in vitro* (**Fig.**

3J). ISCs co-cultured with stromal cells from the L-arginine treated mice generated more organoids (**Fig. 3K**). These data indicated that exogenous L-arginine stimulation of ISC function are mainly mediated through the stromal population. Thanks!

10. In figure 4E, it looks as though IWP-2 alone suppresses organoid formation independent of Arg effects. Can the authors comment on that? Also, based on Fig S2B there may still be some Arg dependent partial effects on Wnt dependent genes.

A: Thanks for your kind question. IWP-2 is a small-molecule antagonist of the Wnt/b-catenin pathway. In figure 4, we found that exogenous L-arginine treatment could active Wnt/ β -catenin pathway *in vivo* and *in vitro*. Therefore, we utilized the Wnt pathway inhibitor IWP-2 to further verify that exogenous L-arginine stimulated ISC-mediated epithelium regeneration through Wnt/ β -catenin pathway. However, among the modulators of ISCs in crypt niches, Wnt/ β -catenin signaling pathway is the most indispensable for ISC expansion and crypt formation⁷. When Wnt/ β -catenin signaling pathway was inhibited, the development of ISCs was stagnant. As you mentioned, IWP-2 alone suppressed organoid formation independent of L-arginine effects. Therefore, after our re-evaluation, we considered the data of IWP-2 was meaningless and decided to remove it from this paper. Previous studies have demonstrated that a number of factors produced by stromal cells have an essential role in the maintenance of ISCs such as Wnt2b⁸; the Lgr4/5 ligand R-spondin1^{9,10,11}; and Gremlin1^{12, 13}. In our study, we found that exogenous L-arginine significantly increased expression of Wnt2b in the CD90⁺ stromals, whereas the levels of other Wnts (Wnt1, Wnt3, Wnt3a, Wnt4, Wnt5a, and Wnt6), Gremlin1, and R-spondin1 did not change (**Fig. 5A**). Although data in Figure 5-7 supported the conclusion that

increased expression of Wnt2b in CD90⁺ stromal cells mediated the effects of L-arginine on ISC function. The microenvironment of ISCs is very complex and there may still be some L-arginine dependent partial effects on Wnt dependent genes. Thanks very much!

11. “However, L-arginine treatment alleviated and protected damage induced by TNF- α on SI organoids to maintain normal morphology”. The protective effects of Arg appear to be somewhat partial here, so the authors should change the conclusions accordingly.

A: Thanks for your kind suggestion. We have changed the conclusions in this part. To further clarify the protective role of L-arginine on intestinal epithelium, we treated intestinal organoids with murine TNF- α to destroy the epithelium as described previously¹⁴. TNF- α successfully induced organoids disruption in co-culture model. The morphology of the organoids was severely damaged with excessive cell death and less budding (**Fig. 7D**). Exogenous L-arginine treatment and murine recombinant WNT2B protein alleviated damage induced by TNF- α and maintained normal morphology of SI organoids (**Fig. 7E**). However, when anti-Wnt2b antibody neutralized the effect of WNT2B protein, L-arginine had no effect on TNF- α -induced SI organoids injury (**Fig. 7E**). Collectively, these data demonstrated that L-arginine supplement and WNT2B protein protected SI organoids from epithelial damage provoked by TNF- α . Thanks!

12. Can the authors elaborate and discuss how their findings fit with previous studies that show pharmacological and dietary mTORC1 suppression has both direct and indirect effects on intestinal stem cells (Yilmaz et al, Igarashi et al). This also goes

back to the question of how L-arginine affects Paneth cell responses.

A: Thanks for the references, which are now included in the revised Discussion section. mTOR is a sensor of nutrients and growth factors. mTORC1 activation promotes cell proliferation by increasing global protein synthesis and other anabolic processes^{15, 16}. mTORC1 signaling has been shown to be required for intestinal epithelium cell proliferation during homeostasis and regeneration^{17, 18, 19, 20}, including regeneration mediated by quiescent intestinal stem cells (ISCs)^{21, 22}. In addition, several studies have shown that calorie restriction (CR) promotes Lgr5⁺ ISC expansion via mTORC1 signaling. In the intestine, CR was reported to increase the number of ISCs by reducing mTORC1 signaling in Paneth or niche cells²³. Unexpectedly, other publication reported that mTORC1 activity in ISCs is up-regulated during CR, resulting in an increase in protein synthesis in ISCs and an increase in their cell number²⁴. Down-regulation of mTORC1 in Paneth cells is necessary for ISC pool expansion, whereas up-regulation of mTORC1 in ISCs is necessary to respond and drive the increase in ISC number. These studies indicated that the regulation of mTORC1 in different cell types is really complicated in the niche of ISC. It might be possible depended on the dose and kind of mTOR agonist or inhibitor, the age and nutritional status of animals and so on. For example, we found low dose of L-arginine supplement (1 mM) could stimulate CD90⁺ stromal cells to secrete Wnt2b and promote Lgr5⁺ ISCs-mediated epithelium renewal. However, high doses of L-arginine supplement (2 mM, 3 mM) destroyed intestinal organoids (**Fig. S2A**). Moreover, exogenous L-arginine stimulation of ISC function are mainly mediated through the stromal population. Paneth cells are not the key cell involved in this biological process. We sincerely hope that our explanation has addressed your question. Thanks!

Minor Comments:

1. In the abstract the authors state that L-Arginine may have a potential as a therapeutic for intestinal regeneration without stating why that may be. Could the

authors add a transition sentence as why they hypothesized L-arginine may have a potential as a therapy for mucosal injury?

A: Thanks for your kind suggestion. We think this sentence is not appropriate in the abstract so we deleted it. We have modified our statement in the abstract. Thanks!

2. Fig S1A is missing units for the x axis.

A: I am sorry for this mistake. We have added the unit of x axis in Figure S1A. Thanks very much for your carefully revision!

3. There are some typos throughout the manuscript that should be fixed.

A: Thanks for your constructive suggestion, which is highly appreciated. We have carefully scrutinized the manuscript, and made corresponding revisions including some typos and grammatical errors.

4. Some of the figures have omitted axis labels and wrong units.

A: I am sorry for these mistakes. We have carefully checked all the figures and made corrections about wrong labels and units. Thanks very much for your carefully revision!

We sincerely hope that this revised manuscript has addressed all your comments and suggestions. We appreciated for your warm work earnestly, and hope that the correction will meet with approval. Once again, thank you very much for your comments and suggestions.

Reviewer #2 (Remarks to the Author):

In this manuscript, Hou et al. investigate the effect of L-arginine on the Lgr5-positive intestinal epithelial stem cell. Previous studies had indicated that L-arginine supplementation was protective against 5-FU injury, but whether this was due to an effect on ISC function and the mechanism of the protection were unclear. The authors found that oral L-arginine supplementation in mice resulted in increased numbers of Lgr5+ stem cells and proliferating Lgr5+ stem cells. This effect was only observed using in vitro organoid culture if CD90+ stromal cells were also present in the culture and was linked to the secretion of Wnt2b from these critical niche cells. The authors then provide data that the altered Wnt2b secretion from the CD90+ stromal cells was due to activated mTOR signaling. Finally, the authors added to existing literature on the protective effect of L-arginine in the 5-FU injury model by demonstrating that increased ISC activity is maintained in L-arginine pre-treated mice. This is overall a very nice study that will be of interest to the community. However, there are areas where the methods or experimental details are unclear, which need to be clarified in order to support the authors' interpretation of these data.

A: Thanks very much for the positive comments and constructive suggestions. Please find the following detailed responses to your comments and suggestions.

Major comments:

What is the rationale for the chosen dose of 7 mg/mL of L-arginine? Also, the control treatment is not stated. Were the control mice given normal drinking water alone? If so, could more specific controls be tested in mice and/or in cultured cells to give some

indication of the specificity of this response to the L-arginine vs. other amino acids?

A: Thanks for your constructive suggestion. Consistent with previous publications^{25, 26}, 8-week-old Lgr5-GFP male mice were orally administrated with L-arginine supplementation (1.5 g/kg body weight/day) for 2 weeks. According to the daily water consumption of mice, we calculated that the concentration of L-arginine added to the drinking water was about 7 mg/ml. To test the specificity of L-arginine's effects in our study, examination the effect of other amino acids is necessary. Moreover, previous studies show that glutamine supplementation could regulate the function of ISCs in mice^{27, 28}. It has long been appreciated that amino acid levels are critical for mTORC1 activation and represent one of the most conserved growth signals to this pathway²⁹. The medium of CD90⁺ stromal cells was supplemented with 1 mM different amino acids (L-arginine, L-leucine, L-lysine, L-glutamine), respectively (**Fig. S4A and B**). We detected the concentration of WNT2B protein in the culture supernatant at day 3 by ELISA. Interestingly, both L-arginine and L-leucine up-regulated the expression of WNT2B protein. However, L-lysine and L-glutamine had no effects. Combining with the data in Figure 5 and 6, these results indicate that the effects of L-arginine on CD90⁺ stromal cells are specific in some degree, which is closely associated with the activation of mTORC1 pathway. However, future studies will need to further explore the specificity of L-arginine's effects on ISCs. Thanks!

How were L-arginine concentrations determined in Fig. 1B? I can't find this information in the manuscript.

A: We are sorry for missing this important information. Thanks very much for your reminding. We utilized the L-Arginine Assay Kit (Abcam, ab241028) to measure total L-arginine concentrations in various samples according to the manufacturer's instructions. We have added "L-arginine measurement" in the **MATERIALS AND METHODS** section. Protocol summary is as below.

1. Prepare small intestinal content, small intestinal tissue, and serum samples, controls and standards as instructed.
2. Prepare the Arginine standard curve.

3. Add 20 μL of sample to desired wells, adjust volume to 40 μL with Arginine Assay Buffer.
4. Prepare the Enzyme mix and add 10 μL to each well containing Standard, Sample, Sample Background and Spiked wells. Prepare the Reaction Mix.
5. Add 200 μL to Standard, Sample, Sample Background and Spiked wells.
6. Measure the absorbance (450 nm) in a microplate in end point mode.

In the images in Fig. 1D, is the green line supposed to indicate what was counted as “crypt”? It appears that the tops of many crypts were not included in the counted region. How was the counted region defined?

A: In figure 1D, we circled the crypt area with white dotted lines. A higher percentage of 5-Ethynyl-2'-deoxyuridine (EdU)-positive cells was observed in small intestinal crypts from mice fed L-arginine. Thanks for your kindly suggestion.

Fig 1E: Please include the full gating strategy for the identification of the Lgr5⁺ cells by flow cytometry. It is not clear how the current gate was drawn, as it appears to be in the middle of the Lgr5 positive-expressing population in the images shown. Also, please follow the journal Reporting Guidelines for flow cytometry – add Lgr5-GFP to the axis label, etc.

A: Thanks for your constructive suggestion. Lgr5⁺ intestinal stem cells were sorted from Lgr5-EGFP-ires-creERT2 (Lgr5-GFP) mice¹. In briefly, single cells were isolated from Lgr5-GFP mice using a modified crypt isolation protocol with 20 min of 5 mM EDTA followed by several strainer steps and a 5-min incubation with TrypLE and 0.8 kU ml⁻¹ DNaseI under minute-to-minute vortexing to make a single-cell suspension. The cell suspensions were stained with 7-AAD (7-amino-actinomycin D) (Biolegends) for flow cytometry analysis to exclude dead cells. Gating for 7-AAD⁻ cells was used to determine and sort Lgr5^{hi} cells. GFP⁺ analysis gates were established so that Lgr5-GFP^{hi} events were detectable in intestinal crypts and organoids from Lgr5-GFP mice (**Fig. 1E and 7B**). Meanwhile, we have elaborated this in the methods. We have included the full gating strategy for the

identification of the live Lgr5^{hi} cells by flow cytometry (**Fig. S1E**) and added Lgr5-GFP to the axis label. Thanks!

The Lgr5-GFP mouse is known to express GFP in a heterogeneous fashion. How was this handled experimentally to generate the data in Fig. 1F? How many crypts were counted? What was done if a crypt had no GFP-expressing cells?

A: Thanks for your kind question. We noticed that the use of Lgr5-GFP mice to count Lgr5⁺ stem cells in intestinal crypts must take into account the mosaicism of GFP expression in the intestines of these mice, which varies from mouse to mouse. If a crypt is completely negative, this may simply be due to its mosaic expression, so only using Lgr5⁺ cell for quantitation is not very accurate. Olfactomedin-4 (Olfm4) can serve as a useful marker for Lgr5-type stem cells in small intestine and colon². Herein, we used OLFM4 antibody (CST, 39141) to stain mouse intestinal stem cells (**Fig. 1F**). We obtained the small intestinal tissues from 3 mice per group and every 20-30 crypts were counted per tissue. We have added both higher and lower magnification of the crypts. This result demonstrated that L-arginine supplementation significantly increased numbers of Olfm4⁺ ISC in small intestinal crypts when compared to control group. Meanwhile, to more accurately analyze the effect of L-arginine on ISCs, we analyzed the number of live Lgr5^{hi} ISC by flow cytometry. Combining these results makes our data more convincing. We sincerely hope that our explanation has addressed your question. Thanks!

The conclusions of Fig 2 could be further supported by including data on the Paneth cells present in organoids +/- L-arginine treatment, as this would be the other key epithelial cell type known to support stem cell function by producing Wnts. Also, did the authors observe any change in mRNA markers for either the Lgr5 stem cell or +4 quiescent intestinal epithelial stem cell?

A: Thanks for your constructive suggestion, which is valuable for improving the accuracy of the manuscript. Previous studies demonstrates that Paneth cells provide

important factors for intestinal stem cell (ISC) maintenance³, such as DLL1/4, TGF- α , Wnt3^{4,5,6}. The enhanced regenerative activity of ISCs in mice fed L-arginine led us to ask whether ISCs responded to L-arginine through the Paneth cell niches. However, exogenous L-Arginine (1 mM) treatment had no effect on numbers of Lysozyme⁺ Paneth cell in organoids (**Fig. S2C**). Similar results were further verified by the mRNA expression of Paneth cell marker genes (*Lyz1*, *Defa6*) (**Fig. S2D**). Moreover, we found that L-arginine supplementation have no significant impact on the mRNA expression of *Wnt2b*, *Wnt3*, *DLL1*, and *DLL4* (**Fig. S2E**). Collectively, these results indicated that the effects of exogenous L-arginine treatment on ISC function might not be mediated through Paneth cells niche. We also tested the mRNA expression of markers for either the Lgr5⁺ ISC (*Lgr5*, *Olfm4*) and +4 quiescent ISC (*Bmi1*, *mTert*) in organoids by RT-PCR (**Fig. S2B**). There is no change between the two groups. Thanks!

Fig 4A,B: If Lgr5-GFP would be on the green channel and b-catenin is also on the green channel, then how was only b-catenin expression quantified in this experiment? Perhaps I am missing something.

A: Thanks for your kind reminding. We have re-done Non-phospho (Active) β -Catenin (Ser45) (CST, 19807) staining in organoids and replaced the Figure 4A with the new image (**Fig. 4A**). We found that the expression of active β -Catenin occurred more in the organoids in co-culture model treated with L-arginine than control group (**Fig. 4A**). Thanks!

Fig 4C: As nuclear b-catenin is a key proliferative signal, these Western blots should be redone using nuclear preps.

A: Thanks for your constructive suggestion. We have detected the expression of Non-phospho (Active) β -Catenin (Ser45) (CST, 19807) in nuclear preps by western

blot. In co-culture model, exogenous L-arginine treatment significantly increased the expression of nuclear Non-phospho (Active) β -Catenin in SI organoids (**Fig. 4B**). However, when CD90⁺ stromals were absent, L-arginine had no effect on the expression of β -Catenin in SI organoid (**Fig. S2A**). Thanks!

Fig 4F: This b-catenin IF seems abnormal. Why is the plasma membrane not staining? It appears that an antibody that generally marks all b-catenin was used, and thus basolateral staining would be expected.

A: Thanks for your kind suggestions, which is valuable for improving the accuracy for the manuscript. We agree with your opinion about β -catenin staining in Figure 4F. We have re-done Non-phospho (Active) β -Catenin (Ser45) (CST, 19807) staining in small intestine and replaced the Figure 4F with the new image (**Fig. 4D**). We found that predominant expression of active β -Catenin occurred more in the small intestine of mice fed L-arginine than control mice (**Fig. 4D**). Thanks!

Can the authors provide any data to suggest that Wnt2b/mTOR activation in CD90⁺ stromal cells are required for the protective effect of L-arginine in the 5-FU *in vivo* experiment?

A: Thanks very much for your comment, which is highly appreciated. To verify Wnt2b/mTOR activation in CD90⁺ stromal cells are required for the protective effect of L-arginine supplementation in the 5-FU *in vivo* experiment, we utilized Wnt2b antibody to neutralize WNT2B protein *in vivo* experiments. Pre-feeding of L-arginine protected mice from intestinal mucosal damage and loss of Lgr5⁺ ISCs due to treatment with 5-FU (**Fig. 7A and B**). Meanwhile, L-arginine treatment resulted in more crypt-formed organoids in the pathological conditions (**Fig. 7C**). These results led us to ask whether the protective effect of L-arginine is mediated through Wnt2b/mTOR activation in CD90⁺ stromal cells. To test this, we neutralized WNT2B

protein *in vivo*. In contrast, treatment of Wnt2b neutralizing antibodies reversed the protective effect of L-arginine on intestinal mucosal damage and loss of Lgr5⁺ ISCs (**Fig. 7A and B**). Meanwhile, Wnt2b neutralizing antibodies inhibited stimulative effect of L-arginine treatment on crypt-formed organoids in mice treated with 5-FU (**Fig. 7C**). In addition, we utilized organoids-stromal cell co-culture model to study the protective effect of L-arginine supplementation on epithelium. TNF- α successfully induced organoids disruption in co-culture model. The morphology of the organoids was severely damaged with excessive cell death and less budding (**Fig. 7D**). Exogenous L-arginine treatment and murine recombinant WNT2B protein alleviated damage induced by TNF- α and maintained normal morphology of organoids (**Fig. 7D and E**). However, when anti-Wnt2b antibody neutralized the effect of WNT2B protein, L-arginine had no effect on TNF- α -induced SI organoids injury. Collectively, these results indicated that L-arginine supplement protects gut from 5-FU and TNF- α induced intestinal epithelial damage in a Wnt2b-dependent manner. Thanks!

Minor comments:

Please indicate the sex of mice used in these experiments.

A: We have indicated the sex of mice used in our experiments. Thanks for your kind suggestion.

Were tests performed to test for parametric vs. non-parametric data distributions to ensure the validity of the 1-way ANOVA and t tests performed in this study?

A: Thanks for your kind question. We have performed test for parametric vs. non-parametric data distributions, which ensured the validity of the 1-way ANOVA

and t tests performed in our study.

Fig 1G: Please mark the figures to indicate what was counted as a live organoid, as there appears to also be dead/non-organoid material in these images.

A: Thanks for your constructive suggestion. As shown in Figure 1G, we have indicated the live organoids by red arrows. Small intestinal crypts from mice fed L-arginine were more likely to form organoid bodies than those from normal fed controls (**Fig.1G**).

In the Reporting Guidelines form, the authors state that a minimum of 3 experiments were performed, but in the manuscript text, they state that a minimum of 2 experiments were performed. Which is it? Please show individual values in graphical data rather than using bar graphs to make this more clear.

A: We are sorry for the wrong statement in the **Reporting Guidelines form**. In our study, a minimum of 2 experiments were performed. The key experiments were performed 3 times. In our manuscript, results are representative of two or three independent experiments. We are sorry for this incorrect statement and made a modification. In addition, we have replaced the bar graphs with dot graphs in the manuscript to make our data more clearly. Thanks for your carefully revisions.

Thanks very much for your careful revisions. We appreciated for your warm work earnestly. We have tried our best to revise the manuscript according to your kind and construction comments and suggestions. We sincerely hope that this revised manuscript has addressed all your comments and suggestions.

References:

1. Snippert HJ, *et al.* Intestinal crypt homeostasis results from neutral competition between symmetrically dividing Lgr5 stem cells. *Cell* **143**, 134-144 (2010).
2. van der Flier LG, Haegebarth A, Stange DE, van de Wetering M, Clevers H. OLFM4 is a robust marker for stem cells in human intestine and marks a subset of colorectal cancer cells. *Gastroenterology* **137**, 15-17 (2009).

3. Sato T, *et al.* Paneth cells constitute the niche for Lgr5 stem cells in intestinal crypts. *Nature* **469**, 415-418 (2011).
4. Gregorieff A, Clevers H. Wnt signaling in the intestinal epithelium: from endoderm to cancer. *Genes Dev* **19**, 877-890 (2005).
5. Ireland H, *et al.* Inducible Cre-mediated control of gene expression in the murine gastrointestinal tract: effect of loss of beta-catenin. *Gastroenterology* **126**, 1236-1246 (2004).
6. Clevers H. Wnt/beta-catenin signaling in development and disease. *Cell* **127**, 469-480 (2006).
7. van der Flier LG, Clevers H. Stem cells, self-renewal, and differentiation in the intestinal epithelium. *Annu Rev Physiol* **71**, 241-260 (2009).
8. Farin HF, Van Es JH, Clevers H. Redundant sources of Wnt regulate intestinal stem cells and promote formation of Paneth cells. *Gastroenterology* **143**, 1518-1529 e1517 (2012).
9. Kim KA, *et al.* Mitogenic influence of human R-spondin1 on the intestinal epithelium. *Science* **309**, 1256-1259 (2005).
10. de Lau W, *et al.* Lgr5 homologues associate with Wnt receptors and mediate R-spondin signalling. *Nature* **476**, 293-297 (2011).
11. Ootani A, *et al.* Sustained in vitro intestinal epithelial culture within a Wnt-dependent stem cell niche. *Nat Med* **15**, 701-706 (2009).
12. Kosinski C, *et al.* Gene expression patterns of human colon tops and basal crypts and BMP antagonists as intestinal stem cell niche factors. *Proc Natl Acad Sci U S A* **104**, 15418-15423 (2007).
13. Hsu DR, Economides AN, Wang X, Eimon PM, Harland RM. The *Xenopus* dorsalizing factor Gremlin identifies a novel family of secreted proteins that antagonize BMP activities. *Mol Cell* **1**, 673-683 (1998).
14. Hou Q, *et al.* Lactobacillus accelerates ISCs regeneration to protect the integrity of intestinal mucosa through activation of STAT3 signaling pathway induced by LPLs secretion of IL-22. *Cell Death Differ* **25**, 1657-1670 (2018).
15. Dowling RJ, *et al.* mTORC1-mediated cell proliferation, but not cell growth, controlled by the 4E-BPs. *Science* **328**, 1172-1176 (2010).
16. Saxton RA, Sabatini DM. mTOR Signaling in Growth, Metabolism, and Disease. *Cell* **169**, 361-371 (2017).

17. Sampson LL, Davis AK, Grogg MW, Zheng Y. mTOR disruption causes intestinal epithelial cell defects and intestinal atrophy postinjury in mice. *FASEB J* **30**, 1263-1275 (2016).
18. Guan Y, *et al.* Repression of Mammalian Target of Rapamycin Complex 1 Inhibits Intestinal Regeneration in Acute Inflammatory Bowel Disease Models. *J Immunol* **195**, 339-346 (2015).
19. Ashton GH, *et al.* Focal adhesion kinase is required for intestinal regeneration and tumorigenesis downstream of Wnt/c-Myc signaling. *Dev Cell* **19**, 259-269 (2010).
20. Haller S, *et al.* mTORC1 Activation during Repeated Regeneration Impairs Somatic Stem Cell Maintenance. *Cell Stem Cell* **21**, 806-818 e805 (2017).
21. Richmond CA, *et al.* Dormant Intestinal Stem Cells Are Regulated by PTEN and Nutritional Status. *Cell Rep* **13**, 2403-2411 (2015).
22. Yousefi M, *et al.* Calorie Restriction Governs Intestinal Epithelial Regeneration through Cell-Autonomous Regulation of mTORC1 in Reserve Stem Cells. *Stem Cell Reports* **10**, 703-711 (2018).
23. Yilmaz OH, *et al.* mTORC1 in the Paneth cell niche couples intestinal stem-cell function to calorie intake. *Nature* **486**, 490-495 (2012).
24. Igarashi M, Guarente L. mTORC1 and SIRT1 Cooperate to Foster Expansion of Gut Adult Stem Cells during Calorie Restriction. *Cell* **166**, 436-450 (2016).
25. Geiger R, *et al.* L-Arginine Modulates T Cell Metabolism and Enhances Survival and Anti-tumor Activity. *Cell* **167**, 829-842 e813 (2016).
26. Sellmann C, *et al.* Oral arginine supplementation protects female mice from the onset of non-alcoholic steatohepatitis. *Amino Acids* **49**, 1215-1225 (2017).
27. Chen S, *et al.* Glutamine supplementation improves intestinal cell proliferation and stem cell differentiation in weanling mice. *Food Nutr Res* **62**, (2018).
28. Moore SR, *et al.* Glutamine and alanyl-glutamine promote crypt expansion and mTOR signaling in murine enteroids. *Am J Physiol Gastrointest Liver Physiol* **308**, G831-839 (2015).
29. Bar-Peled L, Sabatini DM. Regulation of mTORC1 by amino acids. *Trends Cell Biol* **24**, 400-406 (2014).

REVIEWERS' COMMENTS:

Reviewer #1 (Remarks to the Author):

The authors have addressed the majority of my comments and have improved and clarified the manuscript accordingly. One thing to note is that the treatment with L-arginine in vivo does cause an increase in frequency and proliferation not only to ISCs but also Paneth cells. Also, isolated crypts from animals treated with L-arginine had a two-fold increase of clonogenicity in a culturing medium absent of stromal cells. This suggests that either there is some metabolic memory in the stem cells that has already been communicated in vivo and/or signals may still be coming through Paneth cells and/or increased clonogenicity and organoid numbers are due to increased number of stem cells in crypts only. However, the experiment with cultured ISCs alone from Arg treated animals also have an increase in clonogenicity, which argues a somewhat for a niche independent effect (presumably the authors are culturing the exact same number of stem cells from Control and Arg treated animals when they culture the ISCs alone in this assay). That is something the authors should include in the discussion. However, the direct treatment of ISCs with arginine argues against this direct effect of L-arginine on stem cells per say. This is something the authors should definitely discuss further in the discussion. This study is important and adds another layer of complexity in understanding how metabolism can regulate stem cell function and regeneration through the stromal cell compartment.

Reviewer #2 (Remarks to the Author):

The authors' have added substantial clarification and data with this revision, which has greatly improved their manuscript. I have a few very minor remaining comments on this revision that I feel are important to address.

Figure 1: I feel strongly that the authors still must add a clear statement of what the "control" condition is in this figure; I believe it is water, but this question was not answered in the rebuttal. In addition, the authors should add their rationale for choosing this particular dosage of L-arginine to the text. It was obvious from the rebuttal response that this dose was not chosen at random, but rather was well-reasoned – readers will appreciate this information!

Figure 4: Overall, this beta-catenin antibody appears to give a normal staining pattern, so this is a great improvement. In the text, the authors state "We further confirmed in vivo that predominant expression of nuclear localization of β -catenin occurred in the SI crypts of mice fed L-arginine, unlike control mice (Figure 4D)." However, based on the images provided in Figure 4D, it is not possible to discern nuclear staining vs. membrane staining and so these images do not fully back up the authors' statement. Is nuclear beta-catenin staining observed in high magnification images? If so, adding high magnification insert images would be important.

Figure 7: The addition of the Wnt2b-neutralizing antibody data to the in vivo and in vitro experiments in this figure is nice, but neither approach can prove that the Wnt2b is produced by the CD90+ stromal cells. I appreciate that the tools to perform the appropriate cell-specific experiment might not be available, so I do not think that additional experiments are needed. However, it would be important to address the limitation of this experimental approach in the Discussion.

Manuscript: COMMSBIO-20-0325

Title: Exogenous L-arginine increases intestinal stem cell function through CD90+ stromal cells producing mTORC1-induced Wnt2b

Communications Biology

Thanks very much for your and reviews' kind suggestion. We have revised our paper according to reviewers' suggestions. The point to point responses are listed below.

Reviewer #1 (Remarks to the Author):

The authors have addressed the majority of my comments and have improved and clarified the manuscript accordingly. One thing to note is that the treatment with L-arginine in vivo does cause an increase in frequency and proliferation not only to ISCs but also Paneth cells. Also, isolated crypts from animals treated with L-arginine had a two-fold increase of clonogenicity in a culturing medium absent of stromal cells. This suggests that either there is some metabolic memory in the stem cells that has already been communicated in vivo and/or signals may still be coming through Paneth cells and/or increased clonogenicity and organoid numbers are due to increased number of stem cells in crypts only. However, the experiment with cultured ISCs alone from Arg treated animals also have an increase in clonogenicity, which argues a somewhat for a niche independent effect (presumably the authors are culturing the exact same number of stem cells from Control and Arg treated animals when they culture the ISCs alone in this assay). That is something the authors should include in the discussion. However, the direct treatment of ISCs with arginine argues against this direct effect of L-arginine on stem cells per say. This is something the authors should definitely discuss further in the discussion. This study is important and adds another layer of complexity in understanding how metabolism can regulate stem cell function

and regeneration through the stromal cell compartment.

A: Thanks for your kind suggestion. Previous study demonstrated that Wnt signalling induces maturation of Paneth cells in intestinal crypts¹. We found that L-arginine supplement could active Wnt/ β -catenin signals both *in vivo* and *in vitro* (Figure 4). These results partly explained why L-arginine supplement caused an increase in frequency and proliferation not only to ISCs but also Paneth cells. Previous studies demonstrates that Paneth cells provide important factors for ISC maintenance², such as DLL1/4, TGF- α , Wnt3^{3,4,5}. The increase in the number of Paneth cells may in turn affect the function of ISCs. However, exogenous L-arginine treatment had no effect on numbers of Lysozyme⁺ Paneth cell in organoids (Figure S2c). Similar results were further verified by the mRNA expression of Paneth cell marker genes (*Lyz1*, *Defa6*) (Figure S2d). Moreover, we found that L-arginine supplement have no impact on the mRNA expression of *Wnt2b*, *Wnt3*, *Axin2*, and *Ctnnb1* (Figure S2e). Collectively, these results indicated that the effects of exogenous L-arginine treatment on ISC function might not be mediated through Paneth cells niche—or at least not primarily. Both intestinal crypts and ISCs isolated from L-arginine-treated mice were more likely to form organoid bodies than those from the control group (Figure 1g, 1h). These results indicated that increased clonogenicity and organoid numbers are not due to increased number of stem cells in crypts only (the number of ISCs from Control- and Arg-treated mice in this assay was same). L-arginine supplement promoted not only crypts but also ISCs function. However, L-arginine supplement cannot affect the function of ISCs sorted from untreated mice *in vitro*. As you said, the metabolism in stem cell niche is really complex. Perhaps there is some metabolic memory in ISCs sorted from L-arginine-treated mice. Future studies will need to further explore the complexity in understanding how metabolism can regulate stem cell function and regeneration through ISC niche factors. Moreover, we have included these suppose in the Discussion. We sincerely hope that this revised manuscript has addressed all your comments and suggestions. We appreciated for your warm work earnestly, and hope that the correction will meet with approval. Once again, thank you very much for your

comments and suggestions.

Reviewer #2 (Remarks to the Author):

The authors' have added substantial clarification and data with this revision, which has greatly improved their manuscript. I have a few very minor remaining comments on this revision that I feel are important to address.

A: Thanks very much for your careful revision. We have revised the paper according to your suggestions. Please find the following detailed responses to your comments and suggestions.

Figure 1: I feel strongly that the authors still must add a clear statement of what the “control” condition is in this figure; I believe it is water, but this question was not answered in the rebuttal. In addition, the authors should add their rationale for choosing this particular dosage of L-arginine to the text. It was obvious from the rebuttal response that this dose was not chosen at random, but rather was well-reasoned – readers will appreciate this information!

A: We are sorry for missing this important information. As you said, the “control” condition is water without L-arginine supplement. Moreover, we have added this information and the rationale for choosing this particular dosage of L-arginine to the text. Thanks very much!

Figure 4: Overall, this beta-catenin antibody appears to give a normal staining pattern, so this is a great improvement. In the text, the authors state “We further confirmed in vivo that predominant expression of nuclear localization of β -catenin occurred in the SI crypts of mice fed L-arginine, unlike control mice (Figure 4D).” However, based on the images provided in Figure 4D, it is not possible to discern nuclear staining vs. membrane staining and so these images do not fully back up the authors' statement. Is nuclear beta-catenin staining observed in high magnification images? If so, adding

high magnification insert images would be important.

A: Thanks for your kind suggestions, which is valuable for improving the accuracy for the manuscript. We also realized that our description about β -catenin staining in Figure 4D is not very accuracy, so we changed our statement. Now the sentence is “We further confirmed *in vivo* that the increased expression of active β -catenin occurred in the SI crypts of mice fed L-arginine, unlike control mice (Figure 4d)”. Thanks!

Figure 7: The addition of the Wnt2b-neutralizing antibody data to the *in vivo* and *in vitro* experiments in this figure is nice, but neither approach can prove that the Wnt2b is produced by the CD90+ stromal cells. I appreciate that the tools to perform the appropriate cell-specific experiment might not be available, so I do not think that additional experiments are needed. However, it would be important to address the limitation of this experimental approach in the Discussion.

A: We totally agreed with your opinion. We overstated our conclusion in Figure 7. These results only clarified that L-arginine supplement protects gut from 5-FU and TNF- α induced intestinal epithelial damage in a Wnt2b-dependent manner. So we changed our conclusions in Figure 7. Moreover, we addressed the limitation of this experimental approach in the Discussion. Thanks very much!

We sincerely hope that this revised manuscript has addressed all your comments and suggestions. We appreciated for your warm work earnestly, and hope that the correction will meet with approval. Once again, thank you very much for your comments and suggestions. Please allow me to give you my best wishes.

References:

1. van Es JH, *et al.* Wnt signalling induces maturation of Paneth cells in intestinal crypts. *Nat Cell*

Biol **7**, 381-386 (2005).

2. Sato T, *et al.* Paneth cells constitute the niche for Lgr5 stem cells in intestinal crypts. *Nature* **469**, 415-418 (2011).
3. Gregorieff A, Clevers H. Wnt signaling in the intestinal epithelium: from endoderm to cancer. *Genes Dev* **19**, 877-890 (2005).
4. Ireland H, *et al.* Inducible Cre-mediated control of gene expression in the murine gastrointestinal tract: effect of loss of beta-catenin. *Gastroenterology* **126**, 1236-1246 (2004).
5. Clevers H. Wnt/beta-catenin signaling in development and disease. *Cell* **127**, 469-480 (2006).